# Evidence for fungi and gold redox interaction under Earth surface conditions

Tsing Bohu[1], Ravi Anand[1], Ryan Noble[1], Mel Lintern[1], Anna H. Kaksonen[2,3], Yuan Mei [1], Ka Yu Cheng[2,4], Xiao Deng[2], Jean-Pierre Veder[5], Michael Bunce [6], Matthew Power[6] & Mike Verrall[1]

Microbial contribution to gold biogeochemical cycling has been proposed. However, studies have focused primarily on the influence of prokaryotes on gold reduction and precipitation through a detoxification-oriented mechanism. Here we show, fungi, a major driver of mineral bioweathering, can initiate gold oxidation under Earth surface conditions, which is of significance for dissolved gold species formation and distribution. Presence of the gold-oxidizing fungus TA_pink1, an isolate of *Fusarium oxysporum*, suggests fungi have the potential to substantially impact gold biogeochemical cycling. Our data further reveal that indigenous fungal diversity positively correlates with in situ gold concentrations. Hypocreales, the order of the gold-oxidizing fungus, show the highest centrality in the fungal microbiome of the auriferous environment. Therefore, we argue that the redox interaction between fungi and gold is critical and should be considered in gold biogeochemical cycling.

[1] CSIRO Mineral Resources, Australian Resources and Research Centre, Kensington, WA 6151, Australia. [2] CSIRO Land and Water, Private Bag No.5, Wembley, WA 6913, Australia. [3] School of Biomedical Sciences, University of Western Australia, Crawley, WA 6009, Australia. [4] School of Engineering and Information Technology, Murdoch University, Perth, WA 6150, Australia. [5] John de Laeter Centre, Curtin University, Bentley, WA 6102, Australia. [6] Trace and Environmental DNA (TrEnD) Laboratory, Department of Environment and Agriculture, Curtin University, Bentley, WA 6102, Australia. Correspondence and requests for materials should be addressed to T.B. (email: qing.hu@csiro.au)

Gold present in the critical zone originates from dissolved and dispersed primary ores and undergoes a variety of physicochemical interactions[1]. Under Earth surface conditions, gold exhibits enigmatic patterns of transformation and translocation[2,3]. Various forms of gold[4], including secondary aggregates, crystalline grains, colloidal nanoparticles, and Au(I/III) complexes are ubiquitous in diverse environments, such as hypersaline waters[5], iron-bearing nodules[6], and carbonate-rich accumulations[7].

Capillary migration, gaseous transport, and bioturbation are the principal mechanisms hypothetically affect gold speciation, complexation, and mobility in the critical zone[1]. Prokaryotic gold biogeochemical cycling has been proposed and suggests that through bioweathering and oxidative complexation, bacteria and archaea can liberate gold from minerals to form dissolved gold species[8]. Detoxification-oriented biomineralization immobilizes and precipitates dissolved gold species to form secondary deposits[8,9]. Gold transformation subsequently drives changes in the composition of the associated bacterial community[10]. Functional transcriptomic and proteomic analyses of *Cupriavidus metallidurans*, a dominant bacterial species in biofilms on natural gold nuggets, have increased understanding of the role of prokaryotic microorganisms in gold biomineralization[11,12].

Fungi are known to be critical for metals (e.g., aluminum, iron, manganese, calcium, and magnesium) cycling under aerobic Earth surface conditions, influencing essential pedogenic processes, such as rock weathering, soil organic matter degradation, and element distribution[13]. However, little is known about the biogeochemical interaction between fungi and gold. To investigate geomycological contribution to gold cycling in geological records is challenging; partly because it is difficult to empirically verify the fossilized fungi involved in the geological process and distinguish their past activities from those of bacteria or archaea[14]. In addition, fungi are usually better preserved through mineralization than prokaryotes, which further makes it difficult to differentiate between the importance of their respective activities[15]. Recently, fungi have been found to mobilize gold from electronic waste[16], suggesting they can interact with metallic gold, possibly through biological redox transformations.

Gold is extremely resistant to tarnishing (chemical oxidation). Both, an oxidant and a ligand with high affinity for gold ions are required to solubilize gold[17]. Fungi are ideal to simultaneously produce oxidants, e.g., reactive oxygen species[18], and ligands, e.g., organic acids[19], thiosulfate[20], and cyanide[21]. These coordinating molecules also serve as exogenous electron donors to active fungal extracellular oxidation-reduction systems[22], thereby potentially facilitating colloidal gold redox transformation.

Here, we combine X-ray photoelectron spectroscopy (XPS), laser ablation inductively coupled plasma mass spectrometry (LA-ICP-MS), and cyclic voltammetry with geochemical, metagenomic, and geomicrobiological approaches to show that the indigenous fungi in a gold anomaly (Golden Triangle Gold Prospect, Boddington, Western Australia) likely mediate gold oxidation under Earth surface conditions. The existence of an indigenous gold-oxidizing fungus directly links fungal metabolites, processes, and communities with gold biogeochemical cycling.

## Results

### The geology of the gold anomaly
The Golden Triangle Gold Prospect at Zone 50H 442000mE 6376750mN (UTM WGS84) is close to the Boddington Gold Mine, which is located near the edge of the Darling Range, approximately 100 km south east of Perth, Western Australia (Supplementary Fig. 1a). The climate is Mediterranean with an average annual rainfall of 800 mm; mean temperature of 14–32 °C in summer and 4–15 °C in winter. *Eucalyptus* and *Corymbia*-type forests are dominant[23]. In the Golden Triangle Gold Prospect, a 300 m by 100 m secondary gold deposit occurs in the lower part of the aluminum-rich zone and above primary mineralization hosted in volcanic rocks. This secondary gold deposit is within 5–10 m from the surface. A surficial gold anomaly has developed over the mineralization in the near-surface regolith within the iron-rich gravels (Supplementary Fig. 1b). The gold deposit, the overlying soil, and the associated biota (Supplementary Fig. 1c) together constitute an undisturbed geological setting for studying gold distribution and biogeochemical cycling under Earth surface conditions.

### Geochemistry and gold distribution
The surface soil of the Golden Triangle Gold Prospect is mildly acidic and iron-laden (Fig. 1, Supplementary Table 1, and Supplementary Data 1). A comparison between the two areas, including all sampling sites, showed no statistically significant differences in physicochemical parameters except for sulfur content (Fig. 1). In particular, a one-tailed $t$-test showed gold concentrations in the gold anomaly were highly discrete from those in the reference area ($P = 0.07$). Gold was found to be present as <200 nm particles, possibly within the lattice of minerals in the surface soil and distributed heterogeneously in the anomaly to form hotspots. The discrete localized gold concentrations could be up to 40 ng g$^{-1}$, thus elevating the mean gold concentration in the gold anomaly. Sampling sites with in situ gold concentrations 1.5-fold greater than or equal to the median (3.54 ng g$^{-1}$) were determined as gold hotspots in the gold anomaly (Supplementary Fig. 1). Statistically significant differences were detected between the hotspots and the reference area regarding in situ gold concentration ($P = 0.01$); whereas other geochemical parameters, including soil pH, electrical conductivity (Ec), carbon, nitrogen, sulfur, calcium, iron, and water content, were similar (Fig. 1).

### Microbial gold oxidation potential and geochemical modeling
We compared the microbial gold oxidation potential in the gold anomaly and the reference area using six batch-type soil microcosms additionally spiked with 40 µM metallic gold particles (Table 1).

Microcosm GA6F+cs exhibited two phases of linearly increasing oxidation that were staggered and separate from each other (Fig. 2a). The first phase ranged from the initial time point to 45 h and exhibited a relatively high gold oxidation rate of up to 0.19 µM h$^{-1}$. The second phase occurred from 191 to 453 h, at a lower (0.01 µM h$^{-1}$) oxidation rate and followed substantial Au(III) depletion. The concentration of Au(III) shifted linearly with pH (Pearson r = 0.9936, $P = 0.07$) from 45 to 191 h, a rate change period. The concentration of Au(III) in the reference microcosm RA6F+cs increased only between 0 and 45 h, at a rate of 0.07 µM h$^{-1}$, and remained at ca. 5.68 ± 0.59 µM until the end of the experiment (Fig. 2a). No linear relationship between Au(III) and pH (Pearson $r = -0.5184$, $P = 0.65$) was observed from 45 to 191 h. Intriguingly, the Au(III) indicator 3,3′,5,5′-tetramethylbenzidine (TMB, Sigma-Aldrich)[24] revealed no conspicuous Au(III) occurrence in microcosm GA8B+cyc (Fig. 2b), with pH stably around 6. The curves for RA8B+cyc and GA8B+cyc were almost identical. In addition, no unusual pH fluctuations were noted in sterile soil microcosms GA6I and RA6I (Fig. 2c). Concurrently, Au(III) concentration in either microcosm underwent only minimal change, with the exception of an increase (up to 8.48 ± 1.98 µM) in the first 17 h, attributable to a gradual release of loosely bound gold ions from soil matrices and chemical oxidation (e.g., amino acids-mediated oxidation[9]) following soil sterilization[25].

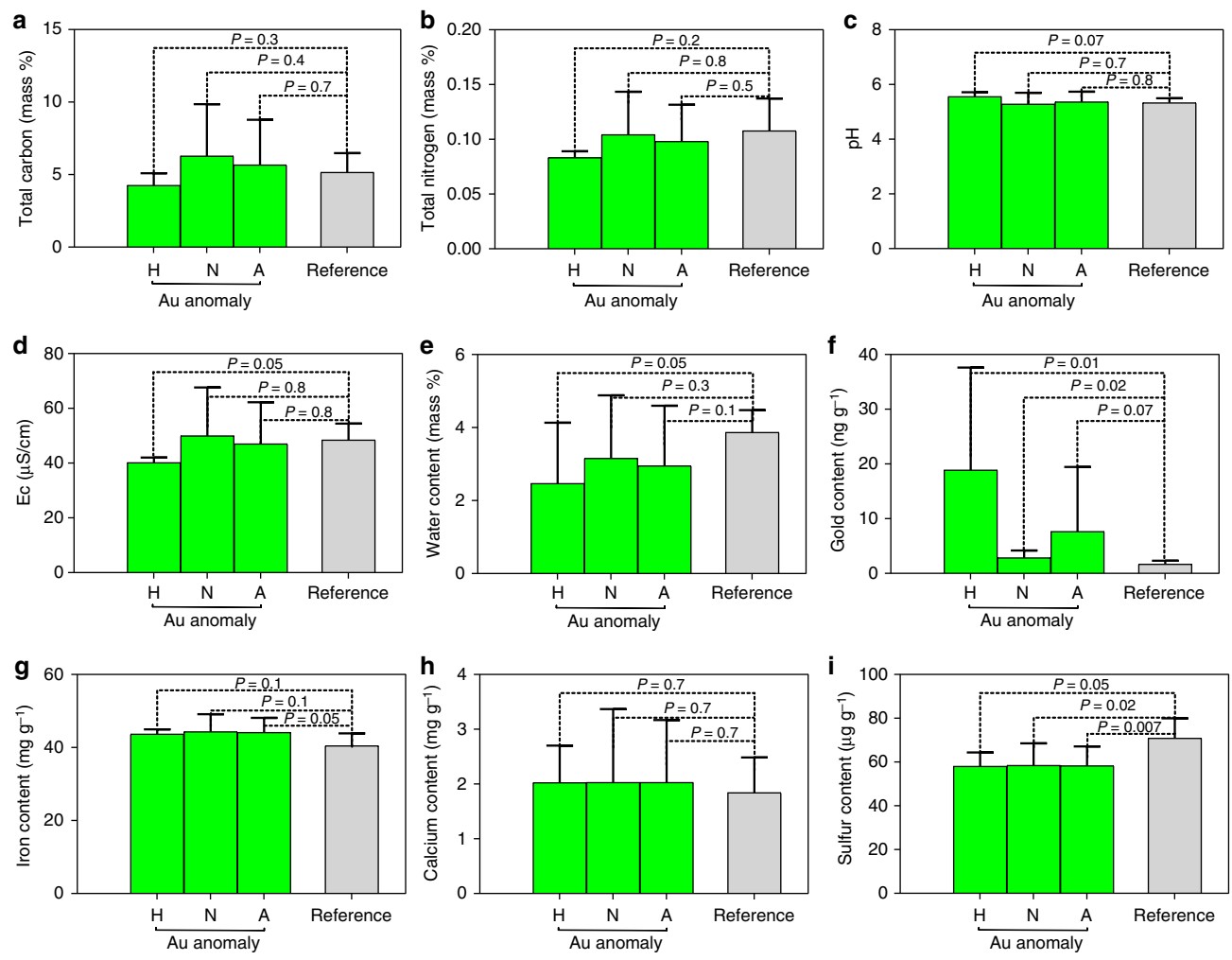

**Fig. 1** Soil geochemical parameters of the gold anomaly and the adjacent reference area. **a–i** represent the values of total carbon, total nitrogen, soil pH, electrical conductivity, water content, the concentrations of gold, iron, calcium, and sulfur. The bars H, N, and A in each panel represent the data of the hotspots, the non-hotspots, and all spots of the gold anomaly, respectively. The significance of the differences between two sets of data was determined by an unpaired $t$-test (GraphPad Prism Version 7). The $P$ value for gold content was based on a one-tailed $t$-test calculation. $P < 0.05$ was considered statistically significant. The error bar indicates the standard deviation

### Table 1 Microcosm conditions

| Microcosms | Sampling sites | | Sampling dates | | Biological activity | | | Antibiotics supplementation | |
|---|---|---|---|---|---|---|---|---|---|
| | Gold anomaly (G) | Reference area (R) | Aug. 2016 (A6) | Aug. 2018 (A8) | Fungi (F) | Bacteria (B) | Inactive (I) | Cycloheximide (+cyc) | Chloramphenicol and streptomycin (+cs) |
| GA6F+cs[a] | ● | | ● | | ● | | | | ● |
| GA8B+cyc | ● | | | ● | | ● | | ● | |
| GA6I | ● | | ● | | | | ● | | |
| RA6F+cs | | ● | ● | | ● | | | | ● |
| RA8B+cyc | | ● | | ● | | ● | | ● | |
| RA6I | | ● | ● | | | | ● | | |

[a]A black dot means the condition was applied to construct the microcosm

We proposed two geochemical models to decipher gold redox transformation in the soil microcosms under carbon-rich and sulfur-rich conditions, respectively. Gold species composition and solubility varied widely depending on pH, redox potential (Eh), and the presence of carbon and sulfur species. In the carbon-rich system (Fig. 2d), an increase in Eh causes the predominant carbon species to change from methane ($CH_4$) to carbon and then to carbonate/bicarbonate. Under oxidizing conditions, gold is transported as hydrated $Au^{3+}$ around pH 0–1 or hydroxyl complex $Au(OH)_4^-$ around pH 1–14. The pH-Eh boundary of the

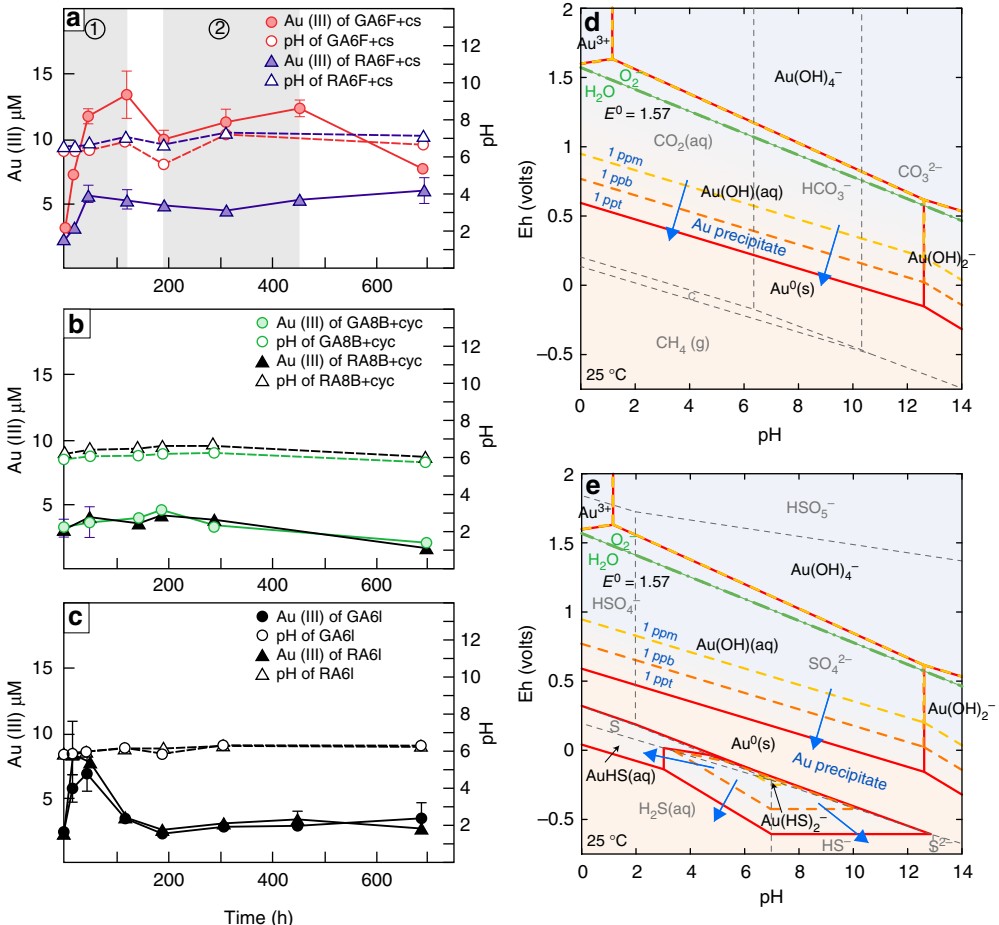

**Fig. 2** Microbial gold oxidation potential and geochemical modeling in soil microcosms. **a–c** Time-dependence between Au(III) concentration and pH in different microcosms expressed as means of triplicates ± standard deviation. **a** Gold oxidation in fungal microcosms GA6F+cs and RA6F+cs, with ① and ② marking the first and second linear phases of gold oxidation in microcosm GA6F+cs. **b** Bacterial microcosms GA8B+cyc and RA8B+cyc. **c** Sterilized microcosms GA6I and RA6I. **d, e** Geochemical models showing predominant Au speciation as a function of pH-Eh at 25 °C. The systems contain initial Au$^+$ activity of $10^{-6}$ to $10^{-12}$, and HCO$_3^-$ activity of 0.01 (**d**) or S$_2$O$_3^{2-}$ activity of 0.01 (**e**). The carbon and sulfur species are reacted with the axis species, and the dashed gray lines show the boundary of predominant carbon and sulfur species over pH-Eh conditions. Predominant Au speciation at 1 part per trillion (ppt) (solid red line), 1 part per billion (ppb) (dashed orange line), and 1 part per million (ppm) (dashed yellow line) is indicated. Blue arrows show the direction of gold mineralization

O$_2^-$/H$_2$O redox couple is close to the boundary of Au(OH)$_4^-$/Au(OH)$_2^-$. With a decrease in Eh, Au$^{3+}$ is reduced to Au$^+$ with the predominant species becoming Au(OH)$_{(aq)}$ and Au(OH)$_2^-$. Under reducing conditions, the predominant Au species is Au$^0_{(s)}$, which is the main mineral form of gold. Organic matter is likely the major chelator in gold mineralization. Under sulfur-rich conditions (Fig. 2e), thiosulfate anion (S$_2$O$_3^{2-}$) is not stable in acidic conditions due to disproportionation. The predominant sulfur species is sulfate (HSO$_4^-$ and SO$_4^{2-}$) under oxidizing conditions and sulfide (H$_2$S and HS$^-$) under reducing conditions. Gold is transported as hydroxyl complexes under oxidizing conditions and Au-HS complexes under reducing conditions. Thus, in both carbon- and sulfur-rich systems, the formation of soluble gold species is affected by the concentration of gold, as well as pH and Eh.

**Isolation and characterization of a gold-oxidizing fungus.** We obtained four gold-oxidizing fungal isolates from a $10^{-1}$ dilution of the gold anomalous soil. One of the isolates belonged to *Fusarium oxysporum* of the phylum Ascomycota and was termed *Fusarium oxysporum* isolate TA_pink1. The sequenced internal transcribed spacer (ITS) region of TA_pink1 was identical (100%)

to that of the model strain *Fusarium oxysporum* CP13, which belongs to the order Hypocreales and has not been reported as a gold-oxidizing fungus. TA_pink1 could grow aerobically on peptone-yeast-glucose (PYG) agar plates[26] with 1200 µM colloidal gold.

When TA_pink1 grew on solid PYG medium supplemented with *ca.* 400 µM colloidal gold for two weeks, a gold depleted halo around the colony appeared, dividing the agar into three parts, a central zone, an oxidized zone, and an undisturbed zone (Fig. 3a). We used LA-ICP-MS to quantify the spatially resolved gold signals on the agar (Fig. 3a, b). The LA-ICP-MS profiles showed gold was enriched in the central fungal biomass ($6.21 \times 10^5 \pm 1.76 \times 10^4$ counts s$^{-1}$, $n = 362$). In contrast, the amount of gold decreased in the oxidized zone ($9.42 \times 10^4 \pm 1.45 \times 10^3$ counts s$^{-1}$, $n = 543$), forming a gold depleted annulus between the central and undisturbed zones ($1.41 \times 10^5 \pm 2.22 \times 10^3$ counts s$^{-1}$, $n = 183$). The normalized trends in gold intensity according to $^{25}$Mg and $^{39}$K (Supplementary Fig. 2) were similar to the gold distribution pattern presented as counts.

We used XPS to determine the distribution of elements and gold species within each zone (Fig. 3c–i). The major peaks noted in the survey spectrum corresponded to O 1s (531.0 eV), C 1s

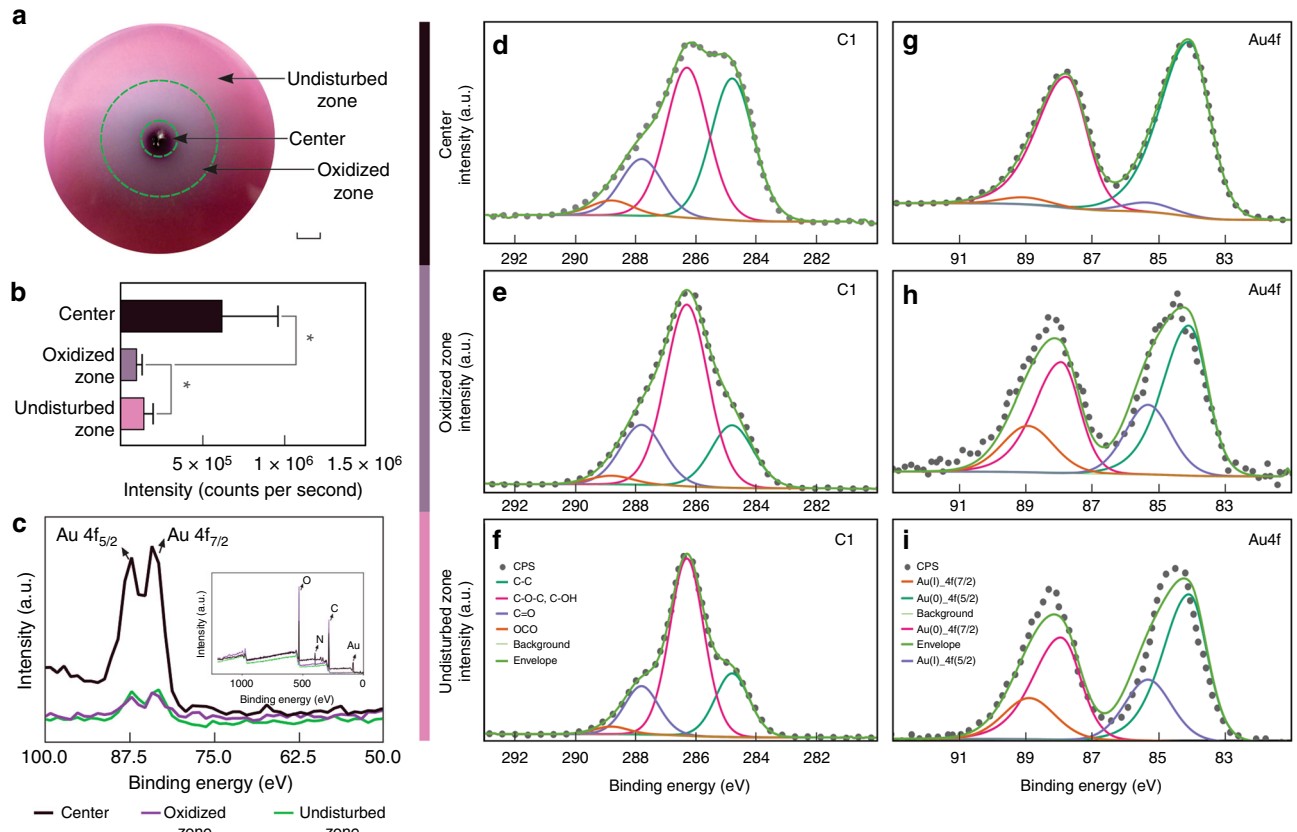

**Fig. 3** Gold-oxidizing capacity of TA_pink1. **a** TA_pink1 was inoculated at the center of PYG agar plates supplemented with 400 µM colloidal gold. After 14 days of incubation at 10 °C in the dark, a gold-dissolving halo appeared around the central colony, dividing the agar into three zones: center, oxidized zone, and undisturbed zone. Scale bar, 1 cm. **b** Quantification of gold intensity in different zones of the PYG agar by LA-ICP-MS. The horizontal axis represents the gold signal in counts s$^{-1}$. Error bars represent standard deviation, *P < 0.001. **c** Expanded gold profile from XPS analysis. The inset shows profiles of major elements, including signals from O 1s (531.0 eV), C 1s (285.1 eV), N 1s (399.5 eV), and Au 4f (84.0 eV). High-resolution scans of C 1s (**d–f**) and Au 4f spectra (**g–i**) from different zones. Gray dots represent the intensity of the spectra in counts per second (cps)

(285.1 eV), N 1s (399.5 eV), and Au 4f (84.0 eV) core levels from species on the agar surface. XPS results were in agreement with LA-ICP-MS profiles. The gold signal in the center was *ca*. 6.5-fold larger than that in the oxidized zone. High-resolution scans of the Au 4f and C 1s regions were deconvoluted, and the parameters extracted from the peaks are shown in Table 2.

Au$^0$ and Au(I) species were present in all three zones, as evidenced from the high-resolution Au 4f spectra, but their relative proportions varied. The oxidized zone had the highest concentration of Au(I) (31.7 At%), followed by the undisturbed (25.2 At%) and central zones (4.0 At%). In addition, the C 1s peak in each of the three zones resolved into four component peaks: C–C from lipids and amino acid side chains; C–O–C or C–OH from alcohols, ethers, and phenols; C=O from carbonyls; and O–C=O from carboxylic acids, carboxylates, and esters[27]. The C=O signal from carbonyl was high in the oxidized zone, whereas O–C=O was enriched in the central zone (Table 2).

We measured cyclic voltammetry of TA_pink1 in the presence and absence of colloidal gold in liquid PYG medium to investigate whether TA_pink1 could stimulate colloidal gold oxidation. If TA_pink1 enhances colloidal gold oxidation, more oxidized Au species, e.g., Au(III) will be present in the reactor and cause higher reducing currents. As shown in Fig. 4a, the initial reducing currents at −0.6 V for TA_pink1 with and without colloidal Au were −0.61 mA and −0.72 mA, respectively. After 17 h incubation of TA_pink1 with colloidal Au, the reducing current increased to −1.01 mA, while it slightly decreased to −0.54 mA when incubated

without Au. Moreover, scanning electron microscopy with energy dispersive spectroscopy (SEM-EDS) revealed that nanometer gold particles occurred on the hyphae of TA_pink1 (Fig. 4b, c) cultured for 14 days in liquid PYG medium with colloidal gold, whereas much larger gold-organic matter complexes were observed in the sterilized control (Supplementary Fig. 3). These differences suggest that TA_pink 1 promoted the oxidation of colloidal Au in liquid PYG medium.

**The ecophysiological effects of gold**. Hyphal extension analysis was carried out to explore the ecophysiological influence of gold on TA_pink1 (Fig. 5, Supplementary Table 2). Sucrose and lignin were chosen as the sole carbon sources in the growth medium because they are common carbohydrate species in plant root exudates[28] and residues[29]. The lag-phase was calculated according to its quantitative definition[30]. TA_pink1 showed a long lag-phase (195.7 h) and drastic colony diameter variation when grown on Czapek Dox agar[31] with sucrose as carbon source (Fig. 5a, Supplementary Table 2). Interestingly, gold supplementation reduced the lag-phase to 38.6 h and attenuated the variation in colony diameter, even though linear extension rates were not altered. By contrast, gold exhibited different effects on hyphal extension when lignin was used as carbon source (Fig. 5b, Supplementary Table 2). Without gold supplementation, the slope of the linear extension period was $0.081 \pm 0.001$ mm h$^{-1}$ and the lag-phase was 83.0 h. Gold slightly accelerated the extension rate ($0.088 \pm 0.0005$ mm h$^{-1}$) and reduced the lag-

**Table 2 High-resolution profiles of Au 4f and C 1s in each agar zone two weeks after inoculation**

| Functional group | Center | | | Oxidized zone | | | Undisturbed zone | | |
|---|---|---|---|---|---|---|---|---|---|
| | Binding energy (eV) | FWHM | At% | Binding energy (eV) | FWHM | At% | Binding energy (eV) | FWHM | At% |
| Au$^0$ | 84.1 | 1.5 | 96.0 | 84.1 | 1.3 | 68.3 | 84.1 | 1.5 | 74.8 |
| Au(I) | 85.4 | 1.5 | 4.0 | 85.3 | 1.6 | 31.7 | 85.4 | 1.6 | 25.2 |
| C–C | 284.8 | 1.7 | 39.0 | 284.8 | 1.7 | 20.0 | 284.8 | 1.3 | 21.4 |
| C–O–C, C–OH | 286.3 | 1.7 | 41.3 | 286.3 | 1.7 | 58.0 | 286.3 | 1.3 | 59.7 |
| C=O | 287.8 | 1.7 | 15.6 | 287.8 | 1.7 | 19.2 | 287.8 | 1.3 | 16.3 |
| O–C=O | 288.8 | 1.7 | 4.1 | 288.8 | 1.7 | 2.8 | 288.8 | 1.3 | 2.6 |

*FWHM* full width at half maximum, *At%* percentage of atoms

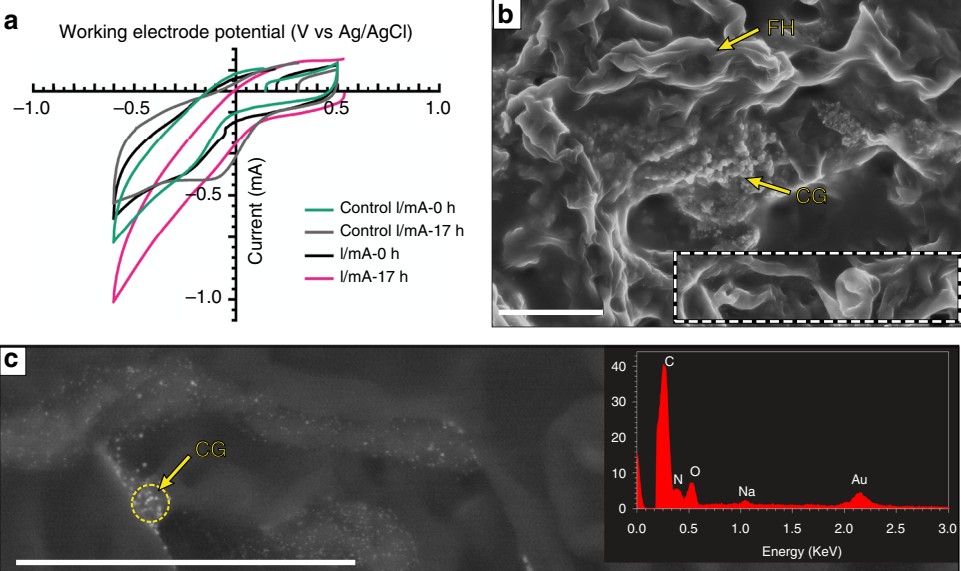

**Fig. 4** Interaction between TA_pink1 and colloidal gold in liquid media. **a** Cyclic voltammograms of TA_pink1 after 0 and 17 h of incubation in the presence and absence of colloidal gold. Control, TA_pink1 incubated without colloidal gold. **b** Scanning electron micrographs of colloidal gold (CG) with the fungal hyphae (FH) of TA_pink1 in liquid PYG medium after 14 days of incubation at 10 °C. Scale bar, 5 μm. **c** Detailed view of nanometer gold particles and surrounding materials (yellow arrows) over the surface of fungal hyphae. Scale bar, 5 μm. The inset shows an EDS profile of the major components of the assemblage

phase (79.9 h), although the patterns of the two extension curves were nearly identical.

**Response of microbial communities to in situ gold**. The composition and structure of indigenous microbial communities were determined by Illumina MiSeq sequencing. Fungal ribosomal RNA (rRNA) gene ITS-targeted sequencing revealed a similar diversity of fungal communities in the gold anomaly and the reference area (Supplementary Tables 3, 4). The inverse Simpson index ($P = 0.7808$, $t = 0.28$, $df = 18$) and the Berger-Parker dominance index ($P = 0.4016$, $t = 0.86$, $df = 18$) showed no statistically significant difference between the two areas. The $t$-test further confirmed no difference in fungal diversity between the hotspots and the reference area (inverse Simpson index, $P = 0.5106$, $t = 0.68$, $df = 11$; Berger-Parker dominance index, $P = 0.6489$, $t = 0.47$, $df = 11$) (Supplementary Table 4). Eight of the ten most abundant fungal operational taxonomic units (OTUs) in the gold anomaly overlapped with those in the reference area. These included Chaetothyriales, Agaricales, Eurotiales, Russulales, Helotiales, Pleosporales, Atheliales, and Capnodiales (Supplementary Fig. 4a). However, the variability in abundance of each fungal OTU differed in the two areas (Supplementary

Fig. 4b, Supplementary Table 3). Pleosporales and Capnodiales were ubiquitous in all samples. Hypocreales were more frequent in the gold anomaly; whereas Chaetothyriales, Eurotiales, and Dothideales were relatively abundant in the reference area.

Bacterial 16S rRNA gene-targeted Illumina MiSeq sequencing showed similarities between the bacterial communities in the gold anomaly and the reference area (Supplementary Table 5). In all sequenced samples, Proteobacteria was the most abundant phylum and accounted for 43.2% of total sequence reads. Actinobacteria represented the second most abundant group, followed by Acidobacteria and Bacteroidetes. The Chao nonparametric richness estimator and inverse Simpson index revealed no statistically significant difference between bacterial diversity in the two areas (Supplementary Table 6).

Intriguingly, Pearson correlation revealed that fungal diversity correlated highly with in situ gold content in the gold anomaly, particularly in the hotspots; whereas no statistically significant correlation was found in the reference area. Bacterial communities seemed not to be influenced by in situ gold content. No statistically significant correlations between in situ gold concentrations and bacterial diversity were detected in either the gold anomaly or in the reference area (Table 3).

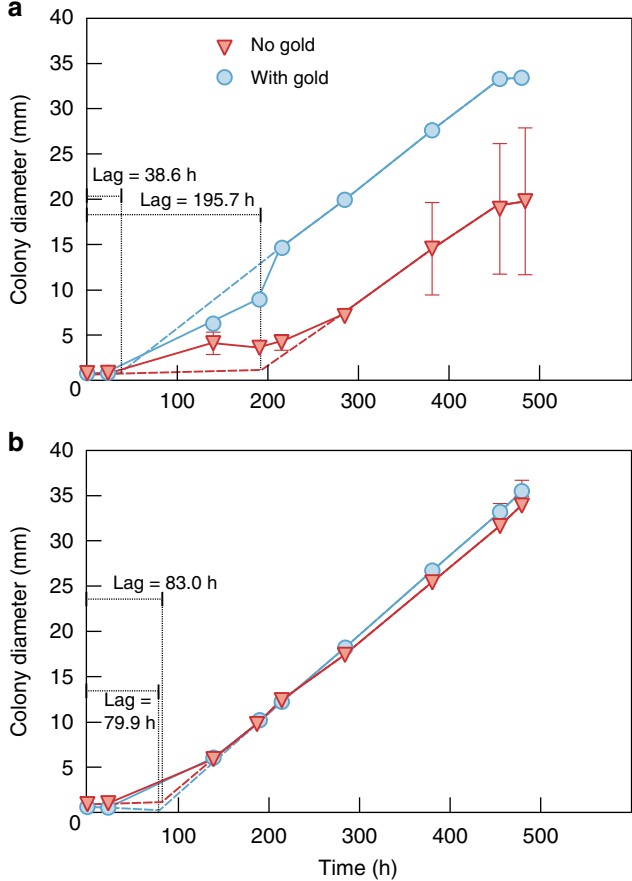

**Fig. 5** Hyphal extension of TA_pink1. (**a**) sucrose and (**b**) lignin as the sole carbon source. Error bars denote the standard deviation of the mean of triplicate measurements and are only shown when greater than the symbol dimension

**Table 3 Pearson correlation between inverse Simpson index and in situ gold concentrations**

| Domain | Pearson correlation | Gold anomaly | Reference |
|---|---|---|---|
| Fungi | $r$ | 0.662 (0.996)[b] | 0.507 |
| | $P$ | 0.037 (0.059) | 0.135 |
| | Significant[a] | +[c] (+) | – |
| Bacteria | $r$ | 0.253 (0.907) | 0.436 |
| | $P$ | 0.511 (0.276) | 0.328 |
| | Significant | –(–) | – |

[a] $P < 0.1$ was considered statistically significant
[b] The values in brackets were from the hotspots
[c] + and – represented significant and non-significant, respectively

Canonical correspondence analysis (CCA) further demonstrated that in situ gold content was an important factor explaining fungal community composition in the gold anomaly, but not in the reference area (Supplementary Fig. 5). Moreover, in situ gold content showed a strong negative correlation with soil pH and intimate association with Hypocreales in the gold anomaly.

Concomitantly, the topology of molecular ecological networks (MENs) showed distinct module-based structural differences between the fungal communities present in the two areas (Fig. 6, Supplementary Data 2–4). Both MENs were divided into three modules using the greedy modularity optimization method[32]. Modules a–c (Fig. 6) represented fungal networks from gold anomalous soils, while modules d–f (Fig. 6) were the reference fungal networks. The MENs' nodes contained seven and eight phyla in the gold anomaly and the reference area, respectively. Although Ascomycota and Basidiomycota were the most dominant phyla in both MENs, the phylogenetic distribution varied substantially among different modules. Module size also varied considerably, ranging from 5 to 26 nodes. Coincident with coefficients of variation (CV) analysis, the MENs of the gold anomaly revealed that Hypocreales (OTU 43) had the highest Stress Centrality $c_S(x)$ (Fig. 6a–c), whereas Atheliales (OTU 55) was the most stressed OTU among reference MENs (Fig. 6d–f). In the gold anomaly, the interactions among the ten most abundant and frequently detected OTUs (determined by 1/CV) were positive and intensive. Seven of the ten OTUs in the gold anomaly clustered in one module (Fig. 6a, Supplementary Table 7). In contrast, competition[33] was the main ecological relationship among the fungal OTUs in the reference area (Fig. 6d–f).

## Discussion

Dissolved gold species (e.g., Au(I/III) complexes) and colloidal gold are commonly found in the critical zone[5,34], and are considered the source of most secondary gold deposits[2]. Given that abiotic formation of ionic gold requires high ionizing energy under Earth surface conditions[35], the interaction between gold and soil biota may be a critical driver of the redox process.

According to the soil microcosm result, indigenous fungi have the potential to markedly influence ionic gold formation in the gold anomaly. In the fungal microcosm GA6F+cs containing the gold anomalous soil spiked with metallic gold (Fig. 2a), the maximum Au(III) concentration occurred at 119 h and corresponded to about 33.6% of the total gold weight in the microcosm, whereas the sterilized control GA6I showed no increase in Au(III) after the initial period (Fig. 2c). The results confirmed that biological redox transformation was the main factor to mediate ionic gold formation in the microcosms. We found that the added metallic gold in microcosms GA6F+cs could not be completely converted to Au(III), indicating that the biological oxidation process was competing with gold reduction and precipitation. Given that the transformation of Au(III) to metallic gold is auto-catalyzed[36], fungal growth phase and metabolism could be critical to gold oxidation. Indeed, Au(III) concentration decreased drastically in microcosm GA6F+cs from 453 h and leveled to that in the reference microcosm, indicating that gold oxidation became attenuated whereas Au(III) reduction and accumulation became dominant in the declining phase. Compared with fungal microcosm GA6F+cs, we detected a weak Au(III) signal in the bacterial microcosm GA8B+cyc containing the gold anomalous soils spiked with metallic gold (Fig. 2b), indicating that the indigenous bacterial consortium had either a low gold oxidation potential or a fast Au(III) turnover. Previous studies revealed that bacteria could rapidly reduce and precipitate Au(III)[37,38]. The rapid reduction and precipitation of Au(III) in GA8B+cyc may hinder colorimetric determination. In fact, a previous study, which did not distinguish between dissolved and colloidal gold, showed that prokaryotic microorganisms could oxidatively dissolve 45.7% (in weight) of gold in soil microcosms after 45 days of incubation[39].

The oxidative dissolution of gold mediated by microorganisms, mainly thiosulfate- and cyanide-producing bacteria[3], is assumed to use molecular oxygen (O₂) as the electron acceptor[40,41]. However, the geochemical models revealed superoxide (O₂⁻) rather than molecular oxygen was a more feasible oxidant for the

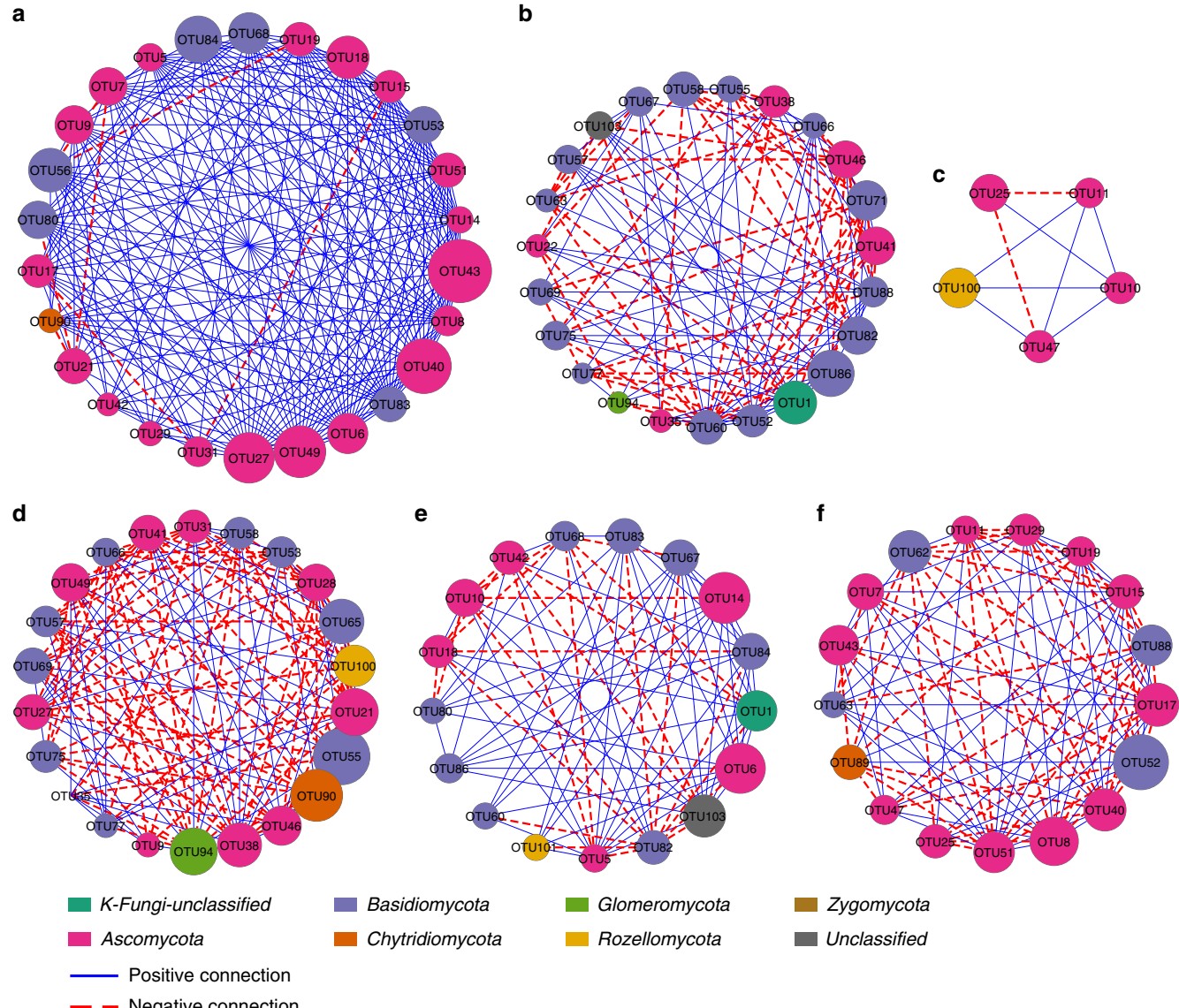

**Fig. 6** Co-occurring ecological networks of fungal OTUs (order level) according to the random matrix theory. Each node represents an OTU. The connectivity between two OTUs is indicated by an edge. Blue and red edges represent positive and negative correlations, respectively. The colors of the nodes denote different phyla. The greedy modularity optimization method was used to form each of the three ecological network modules in the gold anomaly (**a**–**c**) and the reference (**d**–**f**) from fungal communities. The size of each circle indicates the degree of Stress Centrality $c_S(x)$ to each OTU in the gold anomaly and the reference area. The Stress Centrality index is used to describe the approximate amount of stress a node x has to sustain in the network by counting the number of short paths that contain node x. Thus, a node is more central when more short paths run through it. The node at the six o'clock position of each module has the most connections to other nodes. The degree of connection decreases counterclockwise. Detailed taxonomic annotations and the network nodes' centrality indexes for each OTU are shown in Supplementary Data 2–4

generation of Au(III) in carbon-rich and sulfur-rich environments, through the following reactions (Fig. 2d, e):

$$\text{Au}_{(s)} + \text{O}_2^- + 2\,\text{H}_2\text{O}_{(l)} = \text{Au(OH)}_4^- \quad (\log K_{25} = -1.91) \quad (1)$$

$$\text{Au}_{(s)} + 0.75\,\text{O}_{2(aq)} + 2.5\,\text{H}_2\text{O}_{(l)} = \\ \text{Au(OH)}_4^- + \text{H}^+ \quad (\log K_{25} = -16.09) \quad (2)$$

The pH-Eh boundary of the $\text{O}_2^-/\text{H}_2\text{O}$ redox couple is close to the boundary of the $\text{Au(OH)}_4^-/\text{Au(OH)}_2^-$ couple, further supporting the role of superoxide in gold oxidation under Earth surface conditions (Fig. 2d, e). According to reaction (3), gold oxidation is prone to happen at low pH, which is consistent with the correlation of pH and Au(III) concentrations in the rate

change period of microcosm GA6F+cs (Fig. 2a).

$$\text{Au}_{(s)} + \text{O}_2^- + 4\text{H}^+ = \text{Au}^{3+} + 2\,\text{H}_2\text{O}_{(l)} \quad (3)$$

Superoxide is a common microbial metabolic byproduct[18]. Therefore, superoxide-initiated gold oxidation should be a ubiquitous and non-selective process. Indeed, we obtained Au(III) signals in both fungal microcosms, GA6F+cs and RA6F+cs (Fig. 2a). However, relative to GA6F+cs, only a slight linear increase in Au(III) was detected in RA6F+cs between 0 and 45 h (Fig. 2a), indicating that the fungal consortium in the gold anomaly possessed a high gold-oxidizing capacity compared to fungi present in the non-auriferous environment. Considering that high-affinity ligands are required in gold oxidation, it is possible that the indigenous fungi of the gold anomaly secreted gold-specific ligands to stabilize ionic gold in the system. The

ligands are likely to be organic molecules rather than thiosulfate anion ($S_2O_3{}^{2-}$) as suggested by the geochemical models (Fig. 2d, e). LA-ICP-MS and XPS analysis consistently confirmed that *F. oxysporum* isolate TA_pink1 could mobilize gold and excrete carbonyl compounds during the oxidative dissolution of gold (Fig. 3; Table 2), in agreement with a previous finding showing that quinones, a group of carbonyl-rich compounds, were the key molecules in the oxidative transfer of gold[42]. TA_pink1 is thus likely to be an aurophilic microorganism that influences gold speciation via the exudation of a carbonyl-rich ligand with high affinity for ionic gold. In fact, *Fusarium* has been reported to drive stainless steel biofouling by producing a ligand in micro-aerobic and oligotrophic degassing systems of a heavy water ($D_2O$) treatment plant[43]. In addition, the electrochemical reaction between TA_pink1 and colloidal gold demonstrated that TA_pink1could initiate electron transfer via colloidal gold in the liquid phase (Fig. 4a). This process may influence nanometer gold particles formation (Fig. 4b, c) and facilitate oxidative dissolution of gold.

Dissimilar to prokaryotic gold redox transformation, the interaction between fungi and gold is unlikely to involve a detoxification process. Fungal polarized growth under nutrient starvation can induce drastic variations in colony diameter[44]. Au(III) attenuated fungal colony diameter variation and reduced the lag-phase during hyphal extension when sucrose, as opposed to lignin, was the sole carbon source (Fig. 5, Supplementary Table 2). This finding suggests that TA_pink1 prefers to use sucrose rather than lignin when ionic gold is present to accelerate colonization of the environment. Amenabar and colleagues[45] revealed that microbial substrate preference was dictated by differences in the energy required to metabolize the substrate rather than the energy recovered from it. Therefore, it seems that ionic gold can reduce the energy demand for fungi to utilize unfavorable substances.

At community level, fungal diversity within the gold anomaly (the hotspots especially) correlated positively with in situ gold concentrations (Table 3), suggesting that the two variables had a tendency to increase together rather than through toxicity-driven selection[46]. This was not the case for the reference fungal community and bacterial community in either the gold anomaly or in the reference area. Previous studies have indicated that diversity plays a critical role in biomass production[47] and ecosystem stability[48]. Gold, therefore, is very likely a crucial abiotic factor rather than an inactive element in sustaining fungal ecosystems in the auriferous environment. Separately, CCA verified that gold significantly affected the composition of the indigenous fungal community in the gold anomaly (Supplementary Fig. 5). MENs analysis further revealed that fungal interspecific interactions in the gold anomaly and the reference area differed under variable gold concentrations (Fig. 6). Hypocreales (the order of the gold-oxidizing fungus) exhibited the highest centrality in the gold anomaly and intimately associated with other dominant OTUs (Fig. 6a), suggesting that mycological gold redox transformation is likely an important biogeochemical process influencing the indigenous microbiome in the auriferous environment. As a consequence, the redox interaction between fungi and gold can be expected to influence gold distribution in the auriferous environment.

In summary, our study shows that fungi, a major component of the soil microbiome, can mediate gold oxidation under Earth surface conditions. The existence of the gold-oxidizing fungus TA_pink1 suggests fungi are able to substantially impact gold biogeochemical cycling. Fungal metabolites, such as superoxide and carbonyl-rich ligands, plausibly play a role in mycological gold redox transformation (Fig. 7). Unlike in a detoxification-oriented interaction, gold may reduce the energetic constraint for the gold-oxidizing fungus to utilize unfavorable substances and speed up colonization of the surrounding environment. The remarkable centrality of gold-oxidizing fungi underscores mycological gold redox transformation is a critical biogeochemical process for the indigenous microbiome in auriferous environments, which in turn, may profoundly influence gold mobilization and accumulation in the terrestrial ecosystem.

## Methods

**Soil sampling**. We applied a randomized strategy[49] to locate sampling sites in the gold anomaly over the shallow mineralization, as well as sites in an adjacent non-auriferous district 100 m away (serving as the reference) (Supplementary Fig. 1, Supplementary Table 1). Approximately 500 g of surface soil (0–10 cm) from each site were collected. Briefly, a $0.2\,m \times 0.2\,m$ wide $\times 0.4\,m$ deep bulk of soil was excavated using a sterilized spade. Soils were then carefully sampled with a sterilized scoop along the exposed profile. A total of 19 randomly collected soil samples were immediately stored in a $-86\,°C$ portable freezer (ULT25; Global Cooling, Inc., Athens, OH, USA) and a cooler with ice for DNA extraction and culture-dependent analysis, respectively.

**Geochemical characterization**. Soil physicochemical parameters, including temperature, moisture, and electrical conductivity were measured on site with a handheld Thermometer (DT-847U; OneTemp Pty Ltd, Adelaide, SA, Australia) and a HydroSense II Soil Moisture Measurement System (Campbell Scientific Pty Ltd, Garbutt, QND, Australia). Soil pH was determined in deionized water by a pH/EC meter (900-P; TPS Pty Ltd, Brisbane, QND, Australia). Element composition of the soils was analyzed by inductively coupled plasma mass spectrometry (ICP-MS) (Nexion 300Q; Perkin Elmer, Waltham, MA, USA) at LabWest Mineral Analysis Pty Ltd. in Perth, Australia. Detection limits for all elements were at the ng g$^{-1}$ level. Gold concentrations in soil samples were determined by the aqua regia method[23]. Briefly, soil samples were ground to 200 μm, mixed with 100% aqua regia (HCl:HNO$_3$ = 3:1), and digested at room temperature for 24 h. The supernatant obtained by centrifugation was analyzed by ICP-MS. Total carbon and nitrogen were analyzed by CSBP Lab (CSBP Fertilizers, Kwinana, WA, Australia) using the Dumas high temperature combustion method[50].

**Soil microcosms**. Six batch-type microcosms were set up under aerobic conditions with metallic gold particles (1.5–3.0 μm; Sigma-Aldrich; St Luis, MO, USA) supplementation. Conditions for all microcosms are listed in Table 1. Briefly, 20 g wet weight of fresh organic-rich soil was added to sterile 250-mL screw cap Erlenmeyer flasks (PYREX®, Sigma-Aldrich) with 200 mL of sterile Milli-Q water. For microcosms GA6F+cs and RA6F+cs, the mixture was supplemented with 0.2-μm filter-sterilized 30 μg mL$^{-1}$ chloramphenicol (Fisher Scientific, Waltham, MA, USA) and 100 μg mL$^{-1}$ streptomycin sulfate (Fisher Scientific) from ethanolic or aqueous stocks, respectively, to inhibit bacterial growth[51]. For microcosms GA8B+cyc and RA8B+cyc, the mixture was supplemented with 5% cycloheximide (0.2-μm filter-sterilized) to inhibit fungal growth[26]. For microcosms GA6I and RA6I, the mixture was sterilized by repeated autoclaving (121 °C, 20 min, twice over 24 h) to cease biological activities. We added sterilized gold particles to the microcosms to achieve an initial gold concentration of 40 μM. The microcosms were inoculated statically in the dark at 10 °C. Before each sampling, the microcosms were homogenized by gentle shaking. The microcosms GA6F+cs, RA6F+cs, GA6I, and RA6I were sampled at 0, 17, 45, 119, 191, 310, 453, and 693 h; whereas the microcosms GA8B+cyc and RA8B+cyc were sampled at 0, 47.5, 140.5, 185.5, 286.5, and 693 h. We pipetted 1 mL of slurry from each flask to colorimetrically determine the Au(III) concentration using TMB (Sigma-Aldrich) and measured the resulting absorbance at 654 nm[24] with a DR 5000™ UV–Vis spectrophotometer (Hach, Dandenong South, VIC, Australia). The slurry was centrifuged at 3000 × g for 1 min and 0.5 mL supernatant was pipetted into a cuvette (BRAND® standard disposable cuvettes, Sigma-Aldrich). Then, 0.1 mL of TMB indicator was added to the cuvette and mixed with the supernatant by gentle pipetting. The mixture was incubated at 25 °C in the dark for 20 min before the colorimetric measurement. We prepared the TMB indicator by dissolving TMB in a mixture of ethanol (100%, Sigma-Aldrich) and 1.0 M sodium acetate/acetic acid buffer (pH 3.5) at a volume ratio of 4:1. The concentration of TMB in the solution was 2.0 mM. The standard curve was prepared using AuCl$_3$ (Sigma-Aldrich) in Milli-Q water solution at a final concentration of 2, 3, 10, 20, and 30 μM (Supplementary Table 8, Supplementary Fig. 6). The pH of the microcosms was also monitored at the sampling time using a pH meter (MC-80; TPS). The microcosm setup and all measurements were conducted in triplicate.

**Geochemical modeling and thermodynamic calculation**. The pH-Eh diagrams were constructed at 25 °C using the Geochemist's Workbench (GWB), version 12 (Rockware Inc., Golden, CO, USA) to show the predicted gold solubility and predominant gold species. The systems contained 1 ppt to 1 ppm of Au$^+$. HCO$_3{}^-$ and S$_2$O$_3{}^{2-}$ with an activity of 0.01 were used to represent carbon-rich and sulfur-rich systems, respectively. In each diagram (Fig. 2d, e), the carbon or sulfur species were reacted at various pH-Eh conditions, and the predominant carbon or sulfur species in the sub-diagram were then reacted with gold.

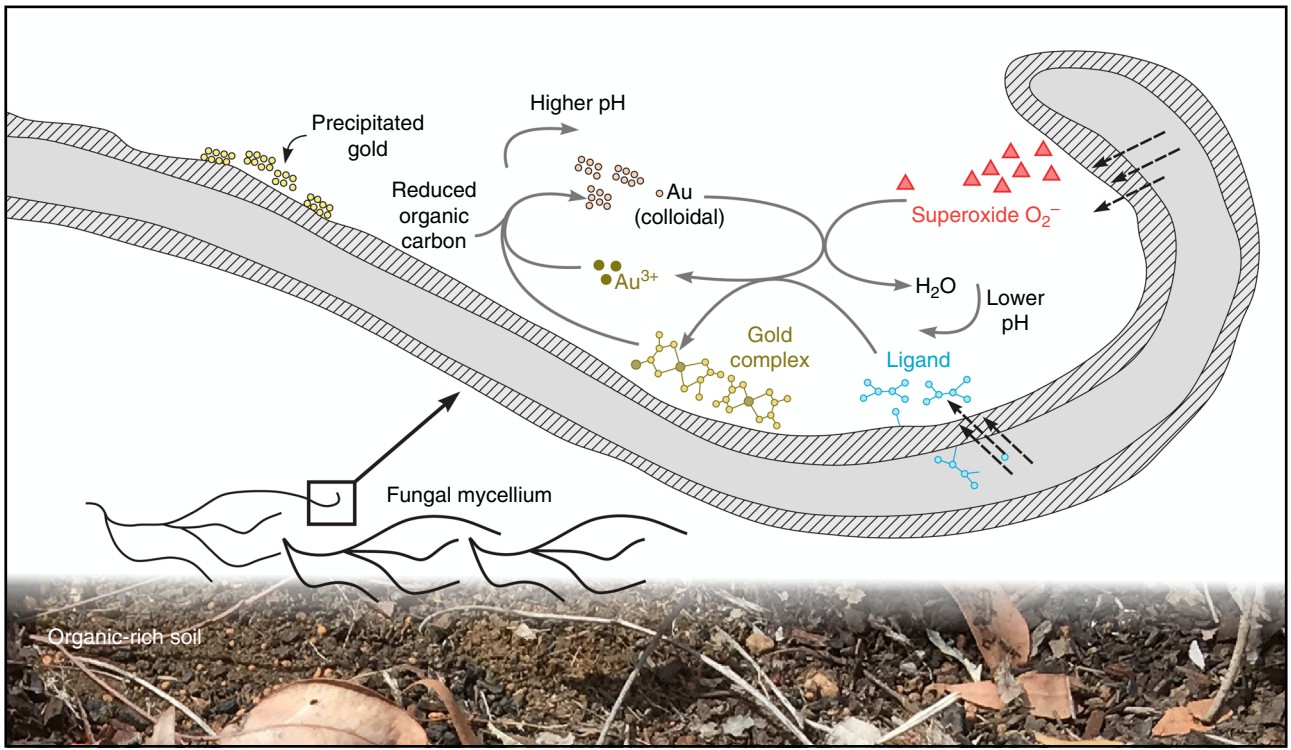

**Fig. 7** The conceptual model for mycological gold redox transformation under Earth surface conditions. Superoxides from fungal hyphae oxidatively dissolve colloidal gold to gold ions with likely assistance of protons. Gold ions then complex with the intracellularly produced ligand. Colloidal gold nanoparticles may be regenerated from the interaction between gold complexes and reduced organic carbon species. Dashed arrows indicate superoxides and ligands are produced intracellularly

Thermodynamic calculation (thermodynamic properties of substances shown in Supplementary Table 9): The standard enthalpy of reaction (3) is:

$$\Delta_r H^\ominus = 2 \times \Delta_f H^\ominus_{H_2O(l)} + \Delta_f H^\ominus_{Au^{3+}} - \Delta_f H^\ominus_{Au^0(s)} - \Delta_f H^\ominus_{O_2^-(g)}$$
$$-4 \times \Delta_f H^\ominus_{H^+} = -122.82 \text{ kJ·mol}^{-1}$$

The standard entropy of reaction (3) is:

$$\Delta_r S^\ominus = 2 \times S^\ominus_{H_2O(l)} + S^\ominus_{Au^{3+}} - S^\ominus_{Au^0(s)} - S^\ominus_{O_2^-(g)} - 4 \times S^\ominus_{H^+} = -359.74 \text{ J·mol}^{-1} \text{ K}^{-1}$$

The Gibbs free energy of reaction (3) is:

$$\Delta_r G^\ominus = \Delta_r H^\ominus - T \times \Delta_r S^\ominus = -15.62 \text{ kJ·mol}^{-1}$$

as $\Delta_r G^\ominus = -2.303 RT \log K^\ominus$ ($R$ is the gas constant, $T$ is temperature in $K$)
The $\log K^\ominus$ of reaction (3) is 2.74.
In alkaline solution[34], $Au^{3+}$ could complex to $OH^-$ forming $Au(OH)_4^-$:

$$Au^{3+} + 4 OH^- = Au(OH)_4^- \quad \log K = 51.35 \qquad (4)$$

Together with water dissociation reaction:

$$H_2O_{(l)} = H^+ + OH^- \quad \log K = -14.00 \qquad (5)$$

By combining reaction (3), (4), and (5), we get the log $K$ of reaction (1)
For reaction of oxygen ($O_{2(aq)}$) oxides $Au_{(s)}^0$

$$Au_{(s)} + 0.75 O_{2(aq)} + 3H^+ = Au^{3+} + 1.5 H_2O_{(l)} \quad \log K = -11.44 \qquad (6)$$

Considering the $Au(OH)_4^-$ complexation, we get the log $K$ of reaction (2) by combining reaction (4), (5), and (6)
To calculate the standard reduction potential of $O_2^-{}_{(g)}/H_2O_{(l)}$ redox couple, the reaction (3) can be divided into two half reactions

$$Au_{(s)} - 3 e^- = Au^{3+} \qquad (7)$$

$$O_2^-{}_{(g)} + 4H^+ + 3 e^- = 2 H_2O_{(l)} \qquad (8)$$

The change in free energy of each half reactions can be measured by Eq. (9)

$$\Delta G^0 = -nFE^0, \qquad (9)$$

where $n$ is number of moles of electrons (equivalents) involved in the reaction; $F$ is the Faraday constant (96.49 kJ per volt gram equivalent), and $E^0$ is the cell potential (V) at standard state. For reaction (7), $E^0 = -1.52V$[52]. So, the $\Delta G^0$

for reaction (7) is:

$$\Delta G^0_{Au^0(s)/Au^{3+}} = -nFE^0 = 440.00 \text{ kJ·mol}^{-1}$$

So, for reaction (8):

$$\Delta G^0_{O_2^-/H_2O(l)} = \Delta_r G^\ominus_{reaction(3)} - \Delta G^0_{Au^0(s)/Au^{3+}} = -455.62 \text{ kJ·mol}^{-1}$$

$$\log K^0 = \frac{-\Delta G^0_{O_2^-/H_2O(l)}}{2.303 \, RT} = 79.84$$

$$E^0_{O_2^-/H_2O(l)} = \frac{-\Delta G^0_{O_2^-/H_2O(l)}}{nF} = 1.57 \text{ V}$$

$$K = \frac{\alpha^2_{H_2O(l)}}{\alpha_{O_2^-} \alpha^4_{H^+} \alpha^3_{e^-}} \qquad (10)$$

So $\log K = 4pH + 3pe$ as $pe = \frac{F}{2.303RT}Eh$
The correlation of pH-Eh for $O_2^-{}_{(g)}/H_2O_{(l)}$ redox couple is

$$79.84 = 4pH + 50.7Eh \qquad (11)$$

As shown in Fig. 2, when pH = 0, Eh = 1.57; when pH = 14, Eh = 0.47.

**Isolation and identification of gold-oxidizing fungi.** The auriferous soil samples were diluted ($10^{-1}$) in sterilized 0.7% NaCl, and 10 μL of the dilution was plated onto gold-PYG agar without adjusting the pH. PYG medium contained (per liter) 20 g agar, 0.25 g peptone, 0.25 g yeast extract, 0.25 g glucose, 0.01 g CaCl$_2$·2H$_2$O, and 0.5 g MgSO$_4$·7H$_2$O[26]. We aseptically added AuCl to the medium to a final concentration of 10 μM. Colloidal gold formed in the medium overnight through AuCl disproportionation. All plates were incubated at 10 °C in the dark. Gold-oxidizing fungi were screened by the TMB plate-flooding method after 14 days of incubation. Briefly, 1 mg of TMB was dissolved in 1 mL of dimethyl sulfoxide (Sigma-Aldrich) and 9 mL of 0.05 M phosphate/citrate buffer (pH 5.0). A drop of TMB reagent was applied directly to a fungal colony and incubated for 20 min at room temperature in the dark prior to a visual inspection for color change. To eliminate false positive signals produced by peroxidase, a drop of 2,2′-azino-bis(3-ethylbenzthiazoline-6-sulfonic acid) (ABTS, Sigma-Aldrich) was applied to a spare area of the same fungal colony. The ABTS solution was prepared by dissolving ABTS in 0.2 M sodium acetate (Sigma-Aldrich) (pH 5.0). The final concentration of ABTS in the solution was 3.6 mM[53]. An additional spot of TMB reagent was placed on the agar as a reference for the abiotic oxidation of gold. TMB-positive (blue) and ABTS-negative (colorless) colonies were

preliminarily identified as gold-oxidizing fungal isolates and were subjected to further analysis after purification by streaking on a plate five times.

Fungal genomic DNA was extracted using a plant/fungi DNA isolation kit (Norgen Biotek Corp., Thorold, ON, Canada) with one loop of fungal biomass, according to the user's manual. Fungal ITS region primers comprising ITS1-F (CTTGGTCATTTAGAGGAAGTAA) (forward) and ITS4 (TCCTCCGCTTATTGATATGC) (reverse) were used for fungal rRNA gene ITS amplification[54]. Sequencing was conducted by the Australian Genome Research Facility (Melbourne, VIC, Australia). Sequences were assembled using Geneious Pro version 4.6.0. Fungal phylogeny was determined by performing a BLAST search of the obtained ITS sequences against the NCBI database.

**X-ray photoelectron spectroscopy.** The preliminarily identified gold-oxidizing fungal isolates were inoculated on gold-PYG agar at a final gold concentration of 400 µM. After 14 days of incubation at 10 °C in dark, a gold-depleted halo around the colony appeared, dividing the agar into a central zone, an oxidized zone, and an undisturbed zone. At the center of each zone, the agar was vertically cut to form a 5 mm wide × 10 mm long surface. The agar pieces were then carefully freeze-dried by lyophilization overnight (ALPHA 2–4 LD plus; John Morris Scientific, Sydney, NSW, Australia). All freeze-dried samples were adhered onto double-sided adhesive tape by exposing the cross-section upside prior to introducing them into the analysis chamber. XPS measurements were performed on an Axis Ultra DLD spectrometer (Kratos, Manchester, UK) using a monochromatic AlKα (1486.6 eV) irradiation source operated at 225 W in conjunction with a charge neutralizer. A hybrid lens system with a magnetic lens provided an analysis area constrained to a spot of 110 µm in diameter. The proportion between area of the beam and sample of each zone was 1:5000. The vacuum pressure of the analysis chamber of the spectrometer was maintained at ≤8 × $10^{-9}$ Torr throughout the duration of the analysis. The electron binding energy scale was referenced to the C 1s line of aliphatic carbon, which was set at 284.8 eV. XPS spectra were collected with a pass energy of 160 eV for survey spectra and 40 eV for high-resolution spectra. Data files were processed using CasaXPS (Casa Software Ltd, Teignmouth, UK). Shirley background subtraction was applied to all high-resolution spectra. Au 4f spectra were fitted using an asymmetric peak shape for metallic Au, whereas Gaussian-Lorentzian line shapes were used to fit the higher-oxidation state gold species. The area ratio for the Au 4f5/2:Au 4f7/2 doublets was set to 3:4, whereas the full width at half maximum was constrained to values considered reasonable for each chemical state.

**Laser ablation inductively coupled plasma mass spectrometry.** Colloidal gold distributions within colonies on the freeze-dried PYG agar plates were measured using LA-ICP-MS. The freeze-dried agar was vertically cut to a 5 mm wide × 80 mm long piece, which was then pressed against a polystyrene plate that had a 2-mm vertical slit to form an even surface for LA-ICP-MS analysis. Profiles (length, ca. 8 cm) covering biomass and agar alone were ablated with a New Wave 193 nm ArF excimer laser coupled to an Agilent 7700 ICP-MS instrument (Agilent Technology, Santa Clara, CA, USA). The line scan ablation was performed using a 30-µm square spot, a laser repetition rate of 20 Hz, and 60 µm s$^{-1}$ stage translation speed. The monitored isotopes were $^{25}$Mg, $^{39}$K, and $^{197}$Au; the dwell times used for these isotopes were 0.01, 0.01, and 0.2 s, respectively. Absolute quantifications of gold were not performed because of the absence of a standard reference. Gold signals from the line scan ablation were presented as counts. Signal intensity was normalized to $^{25}$Mg and $^{39}$K in the agar.

**Cyclic voltammetry.** Cyclic voltammetry was applied to examine gold oxidation by TA_pink1. The cyclic voltammetry experiment was carried out in a 30 mL three-electrode reactor equipped with a graphite rod (diameter 5 mm) working electrode, a Ag/AgCl (3 M KCl) reference electrode, and a platinum wire counter electrode. The electrodes were connected to a potentiostat (SP-150; Bio-Logic, Seyssinet-Pariset, France) via copper wires. Twenty milliliter of PYG medium containing the fungal biomass with or without 400 µM colloidal gold were introduced to the reactor. Cyclic voltammetry was performed at 0 and 17 h at a scan rate of 10 mV/s[55].

**Scanning electron microscopy.** TA_pink1 cultured in 10 mL liquid PYG medium was harvested by centrifugation at 3000 × $g$ for 5 min. Sterilized cotton fiber was used as a non-biological control. Briefly, 200 mL of liquid PYG medium spiked with 400 µM colloidal gold was aseptically mixed with either the TA_pink1 biomass or the sterilized cotton fiber. The mixtures were incubated without shaking at 10 °C in the dark for 14 days. The fungal biomass and cotton fiber were then harvested by centrifugation and freeze-dried by lyophilization overnight (ALPHA 2–4 LD plus). The surface of dried fungal biomass and cotton fiber was coated with carbon and examined with a scanning electron microscope (XL-40 FEG; Philips, Eindhoven, Netherlands) fitted with a semi-quantitative energy dispersive X-ray spectrometer.

**Gold-oxidizing fungus hyphal extension assay.** Czapek Dox agar[31] was sterilized by autoclaving at 121 °C for 20 min prior to addition of 50 µM AuCl$_3$ once the agar had cooled to 50 °C[56]. Then, 30 g of sucrose and 8 g of lignin were added to 1 L of medium, respectively. TA_pink1 was centrally inoculated from actively growing stock cultures maintained on PYG medium and the diameter of the fungal colony was measured manually with a ruler to calculate radial expansion rates.

**Nucleic acid extraction and MiSeq sequencing.** Soil DNA was extracted from 1 g wet weight of soil with a PowerSoil Total DNA Isolation kit (MO BIO Laboratories, Carlsbad, CA, USA) according to the manufacturer's instructions. The extracted DNA was quantified using real-time quantitative polymerase chain reaction (qPCR) to assess template copy number and identify whether PCR inhibitors were present. qPCR was conducted for each sample at three dilutions (undiluted, 10$^{-1}$, and 10$^{-2}$) using the primers ITS2_ITS7F (GTGAGTCATCGAATCTTTG)[57] and ITS2_ITS4R (TCCTCCGCTTATTGATATGC)[58] to sequence the fungal rRNA ITS region, and the primers Bact_16S_F515 (GTGCCAGCMGCCGCGGTAA)[59] and Bact_16S_R806 (GGACTACHVGGGTWTCTAAT)[60] to sequence part of the bacterial 16S rRNA gene.

All PCRs were conducted in a 25-µL volume that included 2.5 mM MgCl$_2$ (Applied Biosystems, Waltham, MA, USA), 1× PCR Gold Buffer (Applied Biosystems), 0.25 mM deoxynucleotide triphosphates (dNTPs) (Astral Scientific, Taren Point, NSW, Australia), 0.4 mg mL$^{-1}$ bovine serum albumin (Fisher Biotec, Wembley, WA, Australia), 0.4 µM of each primer, 0.2 µL AmpliTaq Gold DNA polymerase (Applied Biosystems), and 0.6 µL SYBR-Green dye. The qPCR reaction included denaturation at 95 °C for 5 min, 45 cycles at 95 °C for 30 s, annealing at 54 °C for 30 s, and extension at 72 °C for 30 s; plus a final extension at 72 °C for 10 min. The optimal dilution point for each sample was selected and used as the template for multiplex identifier (MID)-tagged PCR reactions.

All MID-tagged PCRs were performed in a 25-µL volume using the same PCR and thermocycler conditions, as stated above. All amplicons were generated in duplicate and assigned unique forward and reverse MID tags to ensure that any contamination from previously generated amplicons could be excluded post-sequencing. The resulting amplicons were pooled together to reduce PCR stochasticity. They were quantified on a Lab Chip and subsequently pooled in equimolar ratios into an amplicon library. The resulting library was run on a Pippin Prep (Sage Science, Beverly, MA, USA) to size-select for fragments in the range of 250–600 bp to reduce the amount of primer dimers within the library. The resulting eluate was purified using a QIAquick PCR purification kit (Qiagen, Chadstone Center, VIC, Australia) per the manufacturer's protocol. Illumina MiSeq sequencing was performed using the MiSeq Reagent Kit v2 (500 cycles; Illumina, San Diego, CA, USA) 250 bp paired-end protocol following the manufacturer's instructions.

**Statistics and bioinformatics analysis.** Differences and correlation between two sets of data were determined to be significant using an unpaired two-tailed $t$-test and Pearson correlation coefficient (GraphPad Prism Version 7; GraphPad Software, La Jolla, CA, USA).

MiSeq sequencing data were denoised and analyzed using Mothur v1.37.6 on the EC2 cloud service of the Amazon Web Server[61]. According to the MiSeq standard operating procedure (https://www.mothur.org/wiki/MiSeq_SOP), sequences were filtered based on the quality score using the make.contigs command. Sequences were then trimmed to a length of 250 bp, screened for chimeras (UCHIME), and grouped into OTUs at 0.03 distance cut-offs. Statistical analyses and taxonomic classification against the UNITE (fungi)[62] and the Silva (bacteria)[63] reference databases were also performed in Mothur, including the coverage, inverse Simpson index, and Berger-Parker index[64]. Relationships between geochemical variables and fungal community structures were analyzed using CCA in Canoco 5[65].

We constructed MENs based on Illumina MiSeq sequencing data of fungal ITS sequences through random matrix theory[66]. The sequence abundance table of the fungal community after Mothur analysis was split into two datasets: auriferous and reference. For each of them, only the OTUs that appeared in seven or more replicates were used for correlation calculations. The calculations of the global network properties, centrality of individual nodes, and module separation and modularity were performed using Pipeline online (http://ieg2.ou.edu/MENA/main.cgi). Data were visualized by Cytoscape v 2.8.3[67].

## Data availability

The ITS sequence of *F. oxysporum* isolate TA_pink1 was deposited in GenBank under accession number MK156696. The MiSeq sequencing data of the fungal rRNA ITS were deposited in the European Nucleotide Archive under study number PRJEB24170. The MiSeq sequencing data of the bacterial 16S rRNA gene were deposited in GenBank under study number PRJNA504536.

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

## Acknowledgements

This work was supported by a CSIRO Office of Chief Executive postdoctoral fellowship and CSIRO Mineral Resources funding to Tsing Bohu, and CSIRO Environomics Future Science Platform and CSIRO Land and Water funding to Xiao Deng. We thank Graeme Reynolds and Newmont Mining, Ltd. (Newmont) for allowing field sampling work in the Golden Triangle Gold Prospect, Boddington, Western Australia. We thank our colleagues Tania Ibrahimi for drawing the field map and Tenten Pinchand for soil sample preparation. We thank Alex Christ and Masaaki Otsuka from the Bureau Veritas Australia Pty, Ltd. for their assistance with LA-ICP-MS analysis. We thank CSIRO's internal reviewers for their suggestions to improve the manuscript. The authors also acknowledge access to the facilities of the WA X-Ray Surface Analysis Facility, funded by an Australian Research Council LIEF grant (LE120100026).

## Author contributions

T.B., R.A., R.N. and M.L. conceived this study. T.B. designed and executed the experiments with the assistance of R.A., R.N., M.L., A.K., K.Y.C, X.D., J-P.V, M.B., M.P. and M. V. T.B. analyzed the results. Y.M. conducted the geochemical modeling. T.B. wrote the paper. All authors discussed the results and revised the paper.

## Additional information

**Competing interests:** The authors declare no competing interests.

