## [Peer Review File · Nature Communications]

Reviewers' comments:

Reviewer #1 (Remarks to the Author):

"Evidence for geomycological gold redox transformation under supergene conditions" by Bohu and co-authors. I was asked to comment specifically on some of the techniques used in this manuscript (combined X-ray photoelectron spectroscopy, laser ablation inductively coupled plasma mass spectrometry), in addition to the manuscript overall and its interest to specialists in the field of geomycology.

Review:

The manuscript presents an interesting case and uses a broad range of analytical techniques and combine field sampling and lab experiments which is satisfactory. The findings of fungal involvement in gold mobilization should also be of large potential interest. The presentation is unfortunately rather unfocused and although it seems correct that the amount of Au(III) is increased in the gold anomaly sample with fungi, it is in the end not fully clear how the process of gold mobilization and accumulation works and what causes the hotspots of gold in the anomaly. See my comments below.

Comments on techniques:

LA-ICP-MS: This technique relies on normalization to primary and use of secondary standards in order to achieve concentrations. Obviously, for this kind of material there are no suitable standards available and therefore I am pleased to see that the results are reported as counts. However, if the different materials ablate differently, the counts will be affected. In order to correct, or at least show that this is not the case would be to monitor other elements than the element in focus. This has been done for 25Mg and 39K. Uniform counts for these elements should be given for the different zones in order to confirm the trends in Au counts. Please add this to the supplementary information.

XPS: I have no major remarks on this section but I am curious how the samples were measured, as this is a surface sensitive technique and the material shows zonation. Some more information on this should be given in the main text. I presume the beam was aimed at different zones of the agar, but this needs to be given in the text and the proportion of the beam (110 μm) compared to the different zones in the agar needs to be given.

Overall:

I have not looked into details of the sequencing or experimental parts.

Introduction:

Line 41 and onwards: This section about the state of knowledge of fungi in gold cycling could be improved.

Lines 44-45: "For example, fungal fossils are common in the gold-bearing strata of the Witwatersrand goldfield in South Africa that developed in the Archean Eon 14." - 1) The mere finding of fossils of fungi in gold bearing strata does not necessarily confirm involvement of fungi in accumulation of gold, 2) Eukaryotes appeared later in the fossil record. Based on molecular clocks, the fungal clade has been considered to have diverged from remaining opisthokonts around 1.25 Ga (Parfrey, Lahr, Knoll, & Katz, 2011, PNAS). Bengtson et al. (2017, Nature Ecology & Evolution), however, described fossilized fungal-like mycelium-forming organisms in an ophiolite from the Ongeluk formation, South Africa, with an age of 2.4 Ga. These are the oldest eukaryote-like fossils yet described so the Archean fungi cited here from a 1985 publication seems very doubtful.

Line 45: "Anand and colleagues found corroborating evidence of various filamentous fungi-like microbes associated with secondary gold particles and carbonaceous material in Al/Fe-rich pisoliths in the intensely weathered regolith of Yilgarn Craton, Western Australia 15,16" – the citation to Anand and Verrall, 2011 is not correct because there is no mention of gold in that article, merely about fungi. Since the link between gold and fungi in Witwatersrand is doubtful (see my comment above) I would not call it "corroborating" here. Furthermore, in Anand et al 2017, Geology, the specific connection between fungi and gold is hardly discussed and only mentioned at one occasion and in that case it is discussed that "metabolic activities of fungi, bacteria, and other organisms... can account for the formation of

flat-lying Au enrichments in ferricrete..." so the explicit link between fungi and gold deposits in the geological record is not as straightforward as it appears in the introduction.

Lines 94-98, as a geochemist I would like to see the concentrations, e.g. as mean+/-stdev and/or ranges instead, not the least to get a better basic knowledge for the setting.

Line 100: 7.65 ± 3.73 ng/g gold in the anomaly vs 2.04 ± 0.39 in the reference area is not much of an anomaly. Less than twice the concentration if +/- is taken into account. In fact, it seems more likely that there are hotspots in the anomaly area that elevates the mean. Would perhaps be better to use the median here. Also, the box-and-whisker plot should be defined because they are not always used in the same way (especially for mean/median), so what is shown? (range, 25-75% percentiles and mean?). Furthermore, the difference in water content (not statistically significant) between the

areas is on about similar level as gold. So taking the P-value into account there is no statistically significant difference between gold in the anomaly area and the reference area which should be more clearly stated. Gold is not visible in SEM in the samples, so the hotspots cannot be defined in more detail; e.g. whether the gold is contained within mineral phases or it is accessible to biota. Instead of comparing the two areas, wouldn't it be better to compare samples with different gold composition irrespective of area?

The chemical compositions of Au must be much better connected with the fungal communities. In the current version of the manuscript, this connection is not visible. It would be good for instance to show a map of all sampling location with Au concentrations marked (referentially as a contour map based on Au concentrations and on this map, the samples for sequencing and experiments should be marked). Also, Au concentrations in the samples used for the experiment should be given.

Line 213: "Moreover, it seems that pH was a pivotal factor to regulate the patterns of both fungal communities but in opposite trends because an increased pH tended to explain the fungal community structure in the gold anomalous soils, while the distribution of the fungal 215 OTUs was in favor of a decreased pH in the reference area" This sentence needs to be restructured (and caution taken when using PCA alone for interpretation).

Fig 7a: Give scale for the different zones.

Fig 7: legend: "The asterisks indicate $P < 0.001$." Yes but what is the different between * and **?

Fig. 8: Please show the spot size of the EDS analysis on the figure. My experience is that spot size is 5-10 μm which is quite extreme compared to the nano-particles.

Fig. 9: The error bars on sucrose in panel b are very large, and almost make up for the difference with lignin. Explanation for these large error bars is required.

Results in general: There are several sections of discussion (interpretations and comparisons to other works) in the result section. Please move these to the discussion.

Discussion:

Detail: No need to use CZ acronym for critical zone, it is mentioned few times only.

The authors are correct that involvement of prokaryotes has been the focus for biogeochemical cycling of gold and fungi has been less in focus. However, although dealing with accumulation from solution, “Nakajima, 2003, Accumulation of gold by microorganisms, World Journal of Microbiology and Biotechnology, Volume 19, Issue 4, pp 369–374” studied accumulation of gold by bacteria, fungi and yeasts. It was shown that bacteria are more effective to accumulate gold than fungi. Nevertheless the mobilization and oxidation of Au by fungi may indeed be significant and thus worthy of consideration in gold biogeochemical cycling as the authors propose. The authors exclude prokaryotes from their experiments but it would be interesting to explore also a batch where prokaryotes are present, in order to determine whether fungi or prokaryotes have largest gold oxidation potential. Similarly, it would be of large interest also to include sequencing of prokaryotes of the anomaly/reference samples, particularly as the fungi sequences showed small differences between the areas. On the same theme, the potential quantitative difference between prokaryotes and fungi for gold mobilization and accumulation should be discussed.

Line 321: “The result agrees with a previous finding that native gold could be dissolved by resident microbiota in auriferous soils⁴³”. The experimental setup (Fig 9 and related parts) of the current study seems to be quite similar to reference 43, which used prokaryotes. A bit more detailed comparison with that study would be appropriate.

Line 322: “Furthermore, the Au(III) concentration curve in the microcosms of the reference area was similar to that in the sterilized controls” – there is actually an increase in the beginning and then constant values. The anomaly samples show similar increase (but larger) of oxidation potential. Can this be due to the higher Au concentrations of the anomaly sample or are the concentrations of indigenous gold in the sample negligible compared to the added concentrations? There is a need for more information about the nature of the material used in the experiment. After a while, the Au (III) decreases in the anomaly sample, to levels similar to the reference sample. Is this due to accumulation of gold? This needs to be discussed. There is some discussion about it at line 347, but needs to be expanded (and reference given to the figure the discussion is about).

Line 353: “In parallel, the redundancy analysis illustrated that the soil carbon content became a less important restricting factor in shaping the fungal community in the gold anomaly than in the reference area.” In general I am not overly enthusiastic about PCA, and I think it should be used with caution, such as in this case and the following interpretations. Particularly since the two areas showed no significant difference in carbon contents. Therefore, in my view, the conceptual model is not convincingly supported by the data.

Supplementary tables

Please write out all abbreviations at first occasion.

Include Au concentrations of the sequencing samples if possible.

Include chemical composition of all soil samples.

Reviewer #2 (Remarks to the Author):

Review of the manuscript

“Evidence for geomycological gold redox transformation under supergene conditions” by Tsing Bohu with co-authors

The work is related to the geomicrobiological transformations of precious metal gold by fungi and could be of great interest for Nature Communications readers. The manuscript can be published after major revision.

The main strengths of this work are the combination of multidisciplinary approaches to geomycological study of fungal communities of the gold anomaly in the Golden Triangle Gold Prospect including culture-independent and culture-dependent methods as well as the authors' finding that gold may alter the fungal growth on carbon sources (e.g. sucrose) under laboratory conditions.

However, there are following main problems with the current version of the manuscript:

(1) In some places the data interpretation in the text is not very clear and sometimes confusing, for example:

on page 5, the last paragraph regarding the gold, it's not clear whether it was about Fig.2 and if yes there was no significant difference in gold concentration between the anomaly and the reference; there was heterogeneous distribution of gold in the anomaly with site where gold reached 40 ng/g;

talking of ESEM it is not clear what “invisible” gold is: did authors mean that gold content in the sample was below the sensitivity threshold of the EDX analysis? But the gold speciation (e.g. association with clay minerals or soil particles) should not affect the registration of gold as the element in the specimen; the absence of gold in ESEM specimen could be also explained by high heterogeneity of gold distribution;

on pages 6-7, the paragraph with lines 110-126, text regarding the setting of two microcosms is not very clear (here and also on page 20): it could be said simply that there were two kinds of microcosms with (1) reference soil and (2) anomaly soil additionally spiked with gold particles;

also to add the speciation (metallic) of gold particles on line 111, the first time of their mentioning instead of line 121;

on line 124 – are the words about linear relationship related to the correlation analysis pH vs Au(III)? if not it is recommended to check this correlation and to present correlation coefficients and to compare data on current Fig.3, also, somewhere in the paper, it will be very useful to add a COMPREHENSIVE DIAGRAM OF BIOGEOCHEMICAL TRANSFORMATIONS OF GOLD including redox processes, changes in gold speciation, known minerals etc. which will clarify this work, upgrade it and improve its perception by the diverse readers of Nature Communications in general (and will make Fig.3 better understood);

on page 7, line 131, what are the reasons for choosing autoclaving of gold particles over other ways of sterilization (oven-drying, treatment with absolute alcohol etc.) if it could trigger abiotic oxidation, what is the extend of this oxidation, have the authors study it (qualified (changes in speciation) and quantified (how much)) to claim that the alterations of metallic gold was negligible?

on page 11, lines 204-205, it should be “physicochemical”, not “physiochemical”;

on page 12, lines 217-222, to clarify it better (simpler) in the text and to specify parameters (line 220)

on page 12, line 227, what are the reasons of choosing 18S as a phylogenetic marker over universally used ITS?

(2) About fungal growth

on page 15-16, strictly speaking, it was all about the extension of fungal colony on the surface of agar medium; extension is an indicator of colonization but, in many cases, it does not reflect the changes in biomass because the morphology of fungal mycelium can be altered by different growth conditions and a fungus can produce either very thick dense mycelium (phalanx strategy) or, in contrast, long sparsely branched, branchless hyphae (guerrilla strategy); did authors take into account the morphology of fungal mycelium? in case of this paper, the colonization of environment is very important for fungi competing for resources, but it should be specified that it is only an extension (surface growth)

on page 16, Fig. 9, the fungal colony extension is linear, however often there is a delay of surface (or any) growth which is called lag-phase indicating that the microbial culture needs some adaptation to the new environment (the more favourable environment the less the duration of lag-phase);

it is obvious that the fungus TA_pink 1 manifested the lag-phase (extension delay) on sucrose medium on Fig. 9a and the extension rate of linear fungal surface growth on sucrose should be calculated from 284h up to around 500h, and similarly the extension rate calculations of the other data should take into account the lag-phase (growth delay) and be performed ONLY IN THE LINEAR

AREA OF THE PLOT (in current version of the manuscript, there was no point to calculate near-lag-phase areas data and compare them, plus there was no any statistics);

I would strongly recommend the authors to recalculate the extension rates and to make statistical analysis whether there is a significant difference between extension rate of the linear growth on Fig. 9 data as well as to identify the lag-phases durations by the point of cross of the straight line of linear fungal growth with horizontal line parallel to the time axis [see the diagram enclosed, also see Principles of Microbe and Cell Cultivation by S.J. Pirt];

so far, considering that fungus TA_pink 1 was struggling to grow on sucrose under control conditions compared to lignin and looking at the Fig. 9 plots, the main finding is that gold can alter the surface growth of the fungus by reducing its adaptation period (lag-phase duration) to the alternative carbon source sucrose resulting in speeding up the colonization of the environment.

(3) And again, as the title of the manuscript is "Evidence for geomycological gold redox transformation under supergene condition" and the manuscript would greatly benefit from the reinforcing the authors' hypothesis with so called visual summary - a diagram/model and I would strongly recommend making a COMPREHENSIVE DIAGRAM OF BIOGEOCHEMICAL TRANSFORMATIONS OF GOLD BY FUNGI

Reviewer #3 (Remarks to the Author):

General comments:

The manuscript by Bohu and colleges used a combination of molecular, culture-based, and geochemical techniques to identify fungi that can potentially oxidize gold in near surface environments. Using microcosms, they found that gold oxidation occurs more readily than abiotic controls. Furthermore, they were able to obtain a fungal isolate that is capable of oxidizing gold. This is intriguing because gold is typically considered to be biologically inactive. While this manuscript has merit and I see no technical flaws, there are two issues that once fixed would make this a publishable manuscript. First, the discussion over extends the interpretation of the results. None of the methods that were employed are able to deconvolve the mechanism(s) of gold oxidation by this fungus or the ecological interpretation with carbon substrates. Some of this can be in the discussion, but it needs to be toned down and should not be the bulk of the discussion. The discussion should mirror the introduction. Second, readability can be improved. Especially for Nature Communications with a broad readership. Most of my specific comments address the readability issue. I also suggest reducing some of the jargon to help reach a broader audience. For example, supergene means different things if you are a geologist versus a molecular biologist.

Specific comments:

Line 30: split into two sentences

Line 32: What is meant by biological transfer?

Line 34: Awkward sentence that should be removed.

Line 39: change "of" to "on"

Lines 91-94: split into two sentences

Lines 94-98: This exact same information is repeated in figure 2. I suggest stating the actual range of concentrations instead of the statistical parameters.

Line 98: Is this percent water content?

Line 111: Autoclaving was used to sterilize both for the soils and gold. This is typically not the best method because mineralogy can change, carbon can be lost, and often times it is ineffective at sterilization of soil (although spreading the soil out on a large pan seems to help). Some explanation as to why autoclaving was chosen over something like pasteurization and any evidence that it did not affect the outcome of the results would be beneficial.

Line 123: move "may establish" after "dynamic equilibrium"

Line 127: pH is not a great indicator of microbial activity and does not confirm it was sterile.

Lines 151-155: Use standard naming conventions and stick to one taxonomic level when discussing these organisms. The underscores should not be used.

Line 229: Since TA_pink1 has 100% identity, is this really a new strain? Conventional methods to fully name a new strain has not be completed. Until then I would suggest referring to your isolate as “an isolate of *F. verticillioides*”.

Figure 7: Please check the figure contrast for color blind people. Some have difficulty discerning pinks from purples.

Line 264: Is the single asterisks different from the double asterisks.

Line 278: Is it possible that the gold was not reprecipitated, but the imaged gold is what is left from the original colloidal gold that has not dissolved? The difference in minerology has not been determined.

Lines 295 and 299: I suggest providing overall growth rates first and then a brief description of the change that occurs at this time period. This is also something that could be expanded upon in the discussion. Why is there a change in the slope at this time period?

Line 318: With these methods it is not possible to say that fungi were dominant. This was not measured.

Line 325: remove “specific enzymatic or non-enzymatic”

Line 431: Were the same 45 cycles also used? If so this is a lot of cycles that will introduce sequencing errors.

Line 467: “blasting” is not a word. Change to “determined by performing a BLAST search”

Line 523: Was a quality filter also used? This need to be added.

Line 524: Add the percent identity used to create the OTUs.

Response to Reviewer's Comments

We sincerely thank the reviewers for their constructive comments. The revised manuscript has been clarified and strengthened by incorporating these suggestions. Below is a point-to-point response to the specific concerns of each reviewer. We have repeated the reviewer's comments verbatim in italics. The revised content was highlighted in blue color.

Reviewer #1 (Remarks to the Author):

Comment 1: *The manuscript presents an interesting case and uses a broad range of analytical techniques and combine field sampling and lab experiments which is satisfactory. The findings of fungal involvement in gold mobilization should also be of large potential interest. The presentation is unfortunately rather unfocused and although it seems correct that the amount of Au(III) is increased in the gold anomaly sample with fungi, it is in the end not fully clear how the process of gold mobilization and accumulation works and what causes the hotspots of gold in the anomaly. See my comments below.*

Response: We thank reviewer 1 to agree with our paper's main theme and the experimental design. As we mentioned in our manuscript, the biological activity such as bioturbation is merely one of the possible mechanisms that influence gold mobilization and accumulation. To fully address "*how the process of gold mobilization and accumulation works and what causes the hotspots of gold in the anomaly*" would require a much broader paper. Here, we focus on documenting the microbial evidence that fungi are capable of oxidizing gold under Earth surface conditions. We detail how the mutual influence between *in situ* gold and microbiome may enhance our understanding of gold distribution and accumulation from consideration of the perspective of geomicrobiology. Our research is novel in that it incorporates fungi into the gold biogeochemical cycling, which in turn, contributes to the geochemical behavior of gold in the earth. According to Reviewer 1's comments, we have reorganized the results section and refocused

our interpretation of the data.

Comment 2: *LA-ICP-MS: This technique relies on normalization to primary and use of secondary standards in order to achieve concentrations. Obviously, for this kind of material there are no suitable standards available and therefore I am pleased to see that the results are reported as counts. However, if the different materials ablate differently, the counts will be affected. In order to correct, or at least show that this is not the case would be to monitor other elements than the element in focus. This has been done for ^{25}Mg and ^{39}K . Uniform counts for these elements should be given for the different zones in order to confirm the trends in Au counts. Please add this to the supplementary information.*

Response: We thank Reviewer 1's comments on LA-ICP-MS. Indeed, there is no suitable secondary standard to quantify gold concentration in the agar. Although we measured the counts of ^{25}Mg and ^{39}K , they are not ideal and are of limited use to confirm the trends in Au counts in the agar plate. This is because Mg and K are essential nutrition to microorganisms and there will be an enrichment of these elements where the fungal cells are growing, especially in the oxidized zone and the central zone of the agar plate. Nevertheless, we have now normalized the Au counts according to ^{25}Mg and ^{39}K in the agar (Fig. S2). The normalized trends in Au intensity are similar to the Au distribution pattern presented as the counts. We have now adjusted the text to reflect this (Line 169-171).

Line 169-171: The normalized trends in gold intensity according to ^{25}Mg and ^{39}K (Fig. S2) were similar to the gold distribution pattern presented as counts.

Fig. S2

Figure S2. Gold distributions in PYG agar plate, determined using LA-ICP-MS. The intensities were normalized to ^{39}K (a) and ^{25}Mg (b), respectively.

Comment 3: *XPS: I have no major remarks on this section but I am curious how the samples were measured, as this is a surface sensitive technique and the material shows zonation. Some more information on this should be given in the main text. I presume the beam was aimed at different zones of the agar, but this needs to be given in the text and the proportion of the beam (110 μm) compared to the different zones in the agar needs to be given.*

Response: We have now added detailed information about sample preparation according to the reviewer's suggestion (Line 483-490). The ratios between the spot of the beam and the area of each zone were determined (Line 492-494).

Line 483-490: The preliminarily identified gold-oxidizing fungal isolates were inoculated on gold-PYG agar at a final gold concentration of 400 μM . After 14 days of incubation at 10 °C in dark, a gold-depleted halo around the colony appeared, dividing the agar into a central zone, an oxidized zone, and an undisturbed zone. At the center of each zone, the agar was vertically cut to form a 5 mm wide \times 10 mm long surface. The agar pieces were then carefully freeze-dried by lyophilization overnight (ALPHA 2-4 LD plus; John Morris Scientific, Sydney, NSW, Australia). All freeze-dried samples were adhered onto double-sided adhesive tape by exposing the cross-section upside prior to introducing them into the analysis chamber.

Line 492-494: A hybrid lens system with a magnetic lens provided an analysis area constrained to a spot of 110 μm in diameter. The proportion between area of the beam and sample of each zone was 1:5000.

Comment 4: *Line 41 and onwards: This section about the state of knowledge of fungi in gold cycling could be improved.*

Response: Many thanks for the suggestion. We have now revised the section (Line 44-54).

Line 44-54: Fungi are known to be critical for metals (e.g., aluminum, iron, manganese, calcium, and magnesium) cycling under aerobic Earth surface conditions, influencing essential pedogenic processes, such as rock weathering, soil organic matter degradation, and element distribution¹³. However, little is known about the biogeochemical interaction between fungi and gold. To investigate geomycological contribution to gold cycling in geological records is challenging; partly because it is difficult to empirically verify the fossilized fungi involved in the geological process and distinguish their past activities from those of bacteria or archaea¹⁴. The association between carbon seams and certain gold mineralization systems is a distinct feature of geology¹⁵. The carbon seams were originally proposed to represent remnants of eukaryotic microorganisms¹⁶. Recently, fungi have been found to participate in the erosion of a gold sheet¹⁷, suggesting they can interact with metallic gold, possibly through biological

redox transformations.

Comment 5: *Lines 44-45: “For example, fungal fossils are common in the gold-bearing strata of the Witwatersrand goldfield in South Africa that developed in the Archean Eon 14.” - 1) The mere finding of fossils of fungi in gold bearing strata does not necessarily confirm involvement of fungi in accumulation of gold, 2) Eukaryotes appeared later in the fossil record. Based on molecular clocks, the fungal clade has been considered to have diverged from remaining opisthokonts around 1.25 Ga (Parfrey, Lahr, Knoll, & Katz, 2011, PNAS). Bengtson et al. (2017, Nature Ecology & Evolution), however, described fossilized fungal-like mycelium-forming organisms in an ophiolite from the Ongeluk formation, South Africa, with an age of 2.4 Ga. These are the oldest eukaryote-like fossils yet described so the Archean fungi cited here from a 1985 publication seems very doubtful.*

Response: We agree with Reviewer 1 that (1) fungal fossils found in a gold-bearing geological setting do not necessarily confirm the involvement of fungi in gold accumulation; (2) the oldest eukaryote-like fossils appear later than Archean. However, in Peterson & Minski’s 1985 publication¹, they observed not merely fungal fossils in the gold-bearing strata. They found gold accumulated with the fungal fossils intracellularly and extracellularly, which is a promising evidence that the ancient fungi were able to interact with gold and not necessarily at the time of ore formation. In addition, the ancient fungi we mentioned in the first version of our paper is “fossilized Precambrian plant mats formed some 2000 million years ago”. Nevertheless, in view of Reviewer 1’s comments we have now rephrased the section and excluded the debatable information (Line 44-54).

Comment 6: *Line 45: “Anand and colleagues found corroborating evidence of various filamentous fungi-like microbes associated with secondary gold particles and carbonaceous material in Al/Fe-rich pisoliths in the intensely weathered regolith of Yilgarn Craton, Western Australia 15,16” – the citation to Anand and Verrall, 2011 is not correct because there is no mention of gold in that article, merely about*

fungi. Since the link between gold and fungi in Witwatersrand is doubtful (see my comment above) I would not call it “corroborating” here. Furthermore, in Anand et al 2017, Geology, the specific connection between fungi and gold is hardly discussed and only mentioned at one occasion and in that case it is discussed that “metabolic activities of fungi, bacteria, and other organisms.... can account for the formation of flat-lying Au enrichments in ferricrete...” so the explicit link between fungi and gold deposits in the geological record is not as straightforward as it appears in the introduction.

Response: We agree with Reviewer 1. To distinguish the activity between fungi and other microorganisms such as bacteria and archaea in the geological records is challenging. We have now modified the text and deleted the debatable information to reflect this (Line 47-50).

Line 47-50: To investigate geomycological contribution to gold cycling in geological records is challenging; partly because it is difficult to empirically verify the fossilized fungi involved in the geological process and distinguish their past activities from those of bacteria or archaea¹⁴.

Comment 7: *Lines 94-98, as a geochemist I would like to see the concentrations, e.g. as mean+/-stdev and/or ranges instead, not the least to get a better basic knowledge for the setting.*

Response: We have now re-structured the sentences in the paragraph (Line 90-102) and emphasized the geochemical properties of the hotspots (Fig. 2). We have now added the detailed geochemical data in SI (Fig. S1, Table S1 and S2).

Line 90-102: The surface soil of the Golden Triangle Gold Prospect is mildly acidic and iron-laden (Fig. S1, Table S2). A comparison between the two areas, including all sampling sites, showed no statistically significant differences in physicochemical parameters except for sulfur content (Fig. S1; Table S2). In particular, a one-tailed *t*-test showed gold concentrations in the gold anomaly were highly discrete from those in the reference area ($P=0.08$, Fig. S1). Gold was found to be present as <200 nm particles, possibly within the lattice of minerals in the surface soil and distributed heterogeneously in the anomaly to form

hotspots. The discrete localized gold concentrations could be up to 40 ng/g, thus elevating the mean gold concentration in the gold anomaly. Sampling sites NBD02, NBD03, and NBD10 with *in situ* gold concentrations 1.5-fold greater than or equal to the median (3.54 ng/g) were determined as gold hotspots in the gold anomaly (Fig. 1). Statistically significant differences could be detected between the hotspots and the reference area regarding *in situ* gold concentration; whereas other geochemical parameters (e.g., soil pH, electrical conductivity (Ec), carbon, nitrogen, sulfur, calcium, iron, and water content) were similar (Fig. 2).

Fig. 2

Figure 2. Soil geochemical parameters of the hotspots (sites NBD02, NBD03, and NBD10) and the reference area in the Golden Triangle Gold Prospect. The significance of the differences between two sets of data was determined by an unpaired two-tailed *t*-test. $P < 0.05$ was considered statistically significant. The error bar indicates the standard deviation.

Fig. S1

Figure S1. Soil geochemical parameters of the gold anomaly (10 sampling sites, Table S1) and the adjacent reference area. The box extends from the 25th to the 75th percentile. The line in the middle of the box was plotted at the median. The whiskers go down to the smallest value and up to the largest value. The significance of the differences between two sets of data was determined by an unpaired t-test (GraphPad Prism Version 7). The P value for gold content was based on a one-tailed t-test calculation. $P < 0.05$ was considered statistically significant.

Table S1. The coordinates, *in situ* gold concentrations, and experiment design of the samples

Sample ID	Coordinates (UTM WGS84)	Gold concentrations (ng/g)	Geochemical analysis ^a	Microcosm	Gold- oxidizing fungi isolation	MiSeq sequencing (fungal rRNA ITS)	MiSeq sequencing (bacterial 16S rRNA gene)
Gold anomaly							
NBD01	Zone 442121mE 50 H 6376685mN	3.84	●			●	●
NBD02	Zone 442116mE 50 H 6376677mN	40.4	●			●	●
NBD03	Zone 442105mE 50 H 6376675mN	9.91	●	●	●	●	●
NBD04	Zone 442111mE 50 H 6376676mN	3.15	●			●	●
NBD05	Zone 442110mE 50 H 6376671mN	4.70	●			●	●
NBD06	Zone 442102mE 50 H 6376671mN	1.89	●			●	
NBD07	Zone 442111mE 50 H 6376675mN	0.56	●			●	●
NBD08	Zone 442114mE 50 H 6376679mN	3.24	●	●	●	●	●
NBD09	Zone 442118mE 50 H 6376679mN	2.52	●			●	●
NBD10	Zone 442115mE 50 H 6376685mN	6.33	●	●	●	●	●
Reference							
NBD11	Zone 442258mE 50 H 6376791mN	2.22	●			●	●
NBD12	Zone 442255mE 50 H 6376794mN	1.59	●	●		●	●
NBD13	Zone 442246mE 50 H 6376787mN	2.20	●			●	●
NBD15	Zone 442244mE 50 H 6376786mN	1.33	●			●	
NBD16	Zone 442245mE 50 H 6376786mN	2.31	●			●	●
NBD17	Zone 442249mE 50 H 6376799mN	0.51	●	●		●	●
NBD18	Zone 442253mE 50 H 6376798mN	1.73	●	●		●	●
NBD19	Zone 442258mE 50 H 6376798mN	1.13	●			●	
NBD20	Zone 442259mE 50 H 6376797mN	2.20	●			●	

a, a black dot means the sample was applied in the experiment.

Table S2 was shown in a separate Excel form.

Comment 8: Line 100: 7.65 ± 3.73 ng/g gold in the anomaly vs 2.04 ± 0.39 in the reference area is not much of an anomaly. Less than twice the concentration if +/- is taken into account. In fact, it seems more

likely that there are hotspots in the anomaly area that elevates the mean. Would perhaps be better to use the median here. Also, the box-and-whisker plot should be defined because they are not always used in the same way (especially for mean/median), so what is shown? (range, 25-75% percentiles and mean?). Furthermore, the difference in water content (not statistically significant) between the areas is on about similar level as gold. So taking the P-value into account there is no statistically significant difference between gold in the anomaly area and the reference area which should be more clearly stated. Gold is not visible in SEM in the samples, so the hotspots cannot be defined in more detail; e.g. whether the gold is contained within mineral phases or it is accessible to biota. Instead of comparing the two areas, wouldn't it be better to compare samples with different gold composition irrespective of area?

Response: Many thanks for the constructive comments. We agree with Reviewer 1 that there are gold hotspots in the gold anomaly. We have now showed a gold gradient map and identified the gold hotspots in Fig. 1. We compared the geochemical parameters of the hotspots vs. the reference area with much greater differences between Au concentrations (Line 90-102, Fig. 2). Considering the high heterogeneity of the soil samples, we decided to use $P < 0.05$ instead of $P < 0.001$ to represent statistical significance of the geochemical parameters. We have now added more information about the box & whisker plot in the legend and moved the figure (Fig. S1) to SI.

Fig. 1

Figure 1. The geological setting of the secondary gold deposit in the Golden Triangle Gold Prospect, Boddington, Western Australia. (a) Schematic map of sampling sites. The boundary of the buried secondary gold deposit is indicated by a dashed line. Dots represent sampling sites in the gold anomaly and the reference area. Each dot is labeled with a sample ID. The concentration of gold at each sampling site is depicted by a color gradient. The exact gold concentrations and the experimental design are shown in Table S1. Sampling sites NBD02, NBD03, and NBD10 with *in situ* gold concentrations 1.5-fold greater than or equal to the median (3.54 ng/g) were determined as gold hotspots. (b) Dispersion block model of the gold anomaly (after Lintern and Anand, 2017)²⁴. (c) Profile of the surface regolith in the Golden Triangle Gold Prospect. The dark brown layer is organic-rich soil, and the lighter brown layer is loose iron crust. (d) Scanning electron micrograph of abundant fungal mycelia (yellow arrow) in the organic-rich soil.

Comment 9: *The chemical compositions of Au must be much better connected with the fungal communities. In the current version of the manuscript, this connection is not visible.*

Response: To confirm the relationship between microbial (fungal and bacterial) diversity and *in situ* gold concentrations, we have now calculated the Pearson correlation coefficient (Line 258-262, Table 4).

Fungal diversity within the gold anomaly (the hotspots especially) correlated positively with *in situ* gold concentrations. Bacterial communities seemed not to be influenced by *in situ* gold content.

Line 258-262: Intriguingly, Pearson correlation revealed that fungal diversity correlated highly with *in situ* gold content in the gold anomaly, and particularly in the hotspots; whereas no statistically significant correlation was found in the reference area. Bacterial communities seemed not to be influenced by *in situ* gold content. No statistically significant correlations between *in situ* gold concentrations and bacterial diversity were detected in either the gold anomaly or in the reference area (Table 4).

Table 4. Pearson correlation between inverse Simpson index and *in situ* gold concentrations.

Domain	Pearson correlation	Gold anomaly	Reference
Fungi	r	0.662 (0.996) ^b	0.507
	P	0.037 (0.059)	0.135
	Significant ^a	+ ^c (+)	-
Bacteria	r	0.253 (0.907)	0.436
	P	0.511 (0.276)	0.328
	Significant	-(-)	-

a, $P < 0.1$ was considered statistically significant.

b, the values in brackets were from the hotspots.

c, + and - represented significant and non-significant, respectively.

Comment 10: *It would be good for instance to show a map of all sampling location with Au concentrations marked (referentially as a contour map based on Au concentrations and on this map, the samples for sequencing and experiments should be marked). Also, Au concentrations in the samples used for the experiment should be given.*

Response: We tried to add a contour map of *in situ* gold concentrations to Fig. 1. However, due to the heterogeneity of the data, a large area in the map could not be truly contoured. The poorly contoured figure is shown below. We recommend using the new one in the manuscript, which used different color to represent a concentration gradient. Should the reviewer and editor prefer the contoured plot instead we can easily substitute the figure in the manuscript. The hotspots were assigned according to *in situ* gold

concentrations which were 1.5-fold greater than or equal to the median (3.54 ng/g). We added the information in the figure legend. Instead of labelling the samples for sequencing and experiments on the map, the information was now given in Table S1.

Comment 11: Line 213: *“Moreover, it seems that pH was a pivotal factor to regulate the patterns of both fungal communities but in opposite trends because an increased pH tended to explain the fungal community structure in the gold anomalous soils, while the distribution of the fungal OTUs was in favor of a decreased pH in the reference area” This sentence needs to be restructured (and caution taken when using PCA alone for interpretation).*

Response: We have now reanalyzed the pattern of the environmental variables and rephrased the paragraph. We have reduced the emphasis on, and deleted the former context about, pH in the CCA analysis to avoid over interpreting the data, as the reviewer suggested (Line 268-271).

Line 268-271: Canonical correspondence analysis (CCA) further demonstrated that *in situ* gold content was an important factor explaining fungal community composition in the gold anomaly, but not in the reference area (Fig. 8). Moreover, *in situ* gold content showed a strong negative correlation with soil pH and intimate association with *Hypocreales* in the gold anomaly.

Comment 12: Fig 7a: Give scale for the different zones.

Response: We have now added a scale bar which could be used for all the zones (Fig. 4 (former Fig. 7)).

Fig. 4

Figure 4. Gold-oxidizing capacity of TA_pink1. (a) TA_pink1 was inoculated at the center of PYG agar plates supplemented with 400 μM colloidal gold. After 14 days of incubation at 10 °C in the dark, a gold-dissolving halo appeared around the central colony, dividing the agar into three zones: center, oxidized zone, and undisturbed zone. (b) Quantification of gold concentrations in different zones of the PYG agar by LA-ICP-MS. The horizontal axis represents the gold signal in counts/s. Error bars represent standard deviation, * $P < 0.001$. (c) Expanded gold profile from XPS analysis. The inset shows profiles of major elements, including signals from O 1s (531.0 eV), C 1s (285.1 eV), N 1s (399.5 eV), and Au 4f (84.0 eV). High-resolution scans of C 1s (d–f) and Au 4f spectra (g–i) from different zones. Gray dots represent the intensity of the spectra in counts per second (cps).

Comment 13: *Fig 7: legend: “The asterisks indicate $P < 0.001$.” Yes but what is the different between * and **?*

Response: Both P values for * and ** were < 0.001 . Therefore, we changed ** to * to avoid confusion. Please see Fig. 4 (former Fig. 7).

Comment 14: *Fig. 8: Please show the spot size of the EDS analysis on the figure. My experience is that spot size is 5-10 μm which is quite extreme compared to the nano-particles.*

Response: At an accelerating voltage of 7 kV the electron beam is focused to a “spot size” of approximately 1-3 nm and the analytical X-ray excitation volume has a diameter about 100 nm in gold. If the gold particles we are looking at are significantly smaller than 100 nm then most of the electrons will pass through the gold and have an excitation volume in the carbon with a diameter of about 400 nm. What we are showing in Fig. 5 (former Fig. 8) is the spectrum from a roughly spherical volume with a diameter of about 400 nm.

Fig. 5

Figure 5. Interaction between TA_pink1 and colloidal gold in liquid media. (a) Cyclic voltammograms with a three-electrode electrochemical chamber for the same mixture containing both colloidal gold and fungal biomass after 0 and 17 h of incubation. The control was pure fungal biomass without colloidal gold supplementation. (b) Scanning electron micrographs of colloidal gold (CG) with the fungal hyphae (FH) of TA_pink1 in liquid PYG medium after 14 days of incubation at 10 °C. (c) Detailed view of nanometer gold particles (yellow arrows) over the surface of fungal hyphae. The inset shows an EDS profile of the major components of the secondary nanometer gold particles.

Comment 15: *Fig. 9: The error bars on sucrose in panel b are very large, and almost make up for the difference with lignin. Explanation for these large error bars is required.*

Response: We believe that Reviewer 1 was referring to the error bars on sucrose in panel a but not in panel b. The error bar represents the standard deviation of the colony diameter. We found among the three

replicated agar plates there was a plate where the fungal colony diameter was much shorter than others on sucrose without gold amendment. We think the heterogeneity of fungal colony diameter among the replicates is due to fungal polarized growth under nutrition starvation. Yuka Kayano et al.² reported in 2013 that fungal mutants of genes that regulate hyphal growth show variable hyphal diameters under nutrient limited conditions, indicating nutrients availability is an external factor to control hyphal variability. Relative to lignin, sucrose is an unfavorable carbon substance to the gold-oxidizing fungus TA_pink1. Therefore, TA_pink1 developed a variable hyphal diameter when grown on sucrose but not the case when TA_pink1 was grown on lignin. Nevertheless, although the fungal colony diameter varied, the linear regression analysis showed no statistically significant difference of the linear extension rates between the treatments with and without gold on sucrose (Fig. 6, Table 3). We have now discussed the results (Line 364-372).

Fig. 6

Figure 6. Hyphal extension of TA_pink1 with (a) sucrose and (b) lignin as the sole carbon source. Error bars denote the standard deviation of the mean of triplicate measurements and are only shown when greater than the symbol dimension.

Table 3. Linear regression analysis of hyphal extension

	Sucrose		Lignin	
	No gold	With gold	No gold	With gold
Best-fit values ± SE				
Slope	0.070±0.005	0.078±0.0005	0.081±0.001	0.088±0.0005
Y-intercept	-12.38±1.9	-2.05±0.2	-5.45±0.5	-6.73±0.2
X-intercept	177.90	26.36	66.96	76.67
Goodness of Fit				
R square	0.9947	0.9999	0.9990	0.9999
Equation	Y = 0.07×X - 12.4	Y = 0.078×X - 2.1	Y = 0.081×X - 5.5	Y = 0.088×X - 6.7
Differences of the linear extension rates				
P value		0.10		0.0062
Significance (P < 0.01)		NO		YES
Lag-phase (h)	0-195.7	0-38.6	0-83.0	0-79.9

Line 364-372: Dissimilar to prokaryotic gold redox transformation, the interaction between fungi and gold is unlikely to involve a detoxification process. Fungal polarized growth under nutrient starvation can induce drastic variations in colony diameter⁴⁷. Au(III) attenuated fungal colony diameter variation and reduced the lag-phase during hyphal extension when sucrose, as opposed to lignin, was the sole carbon source (Fig. 6, Table 3). This finding suggests that TA_pink1 prefers to use sucrose rather than lignin when ionic gold is present to accelerate colonization of the environment. Amenabar and co-workers⁴⁸ revealed that microbial substrate preference was dictated by differences in the energy required to metabolize the substrate rather than the energy recovered from it. Therefore, it seems that ionic gold can reduce the energy demand for fungi to utilize unfavorable substances.

Comment 16: *There are several sections of discussion (interpretations and comparisons to other works) in the result section. Please move these to the discussion.*

Response: We have now moved these sections to the discussion as suggested.

Comment 17: *No need to use CZ acronym for critical zone, it is mentioned few times only.*

Response: We have now changed all “CZ” to “Critical Zone”.

Comment 18: *The authors are correct that involvement of prokaryotes has been the focus for biogeochemical cycling of gold and fungi has been less in focus. However, although dealing with accumulation from solution, “Nakajima, 2003, Accumulation of gold by microorganisms, World Journal of Microbiology and Biotechnology, Volume 19, Issue 4, pp 369–374” studied accumulation of gold by bacteria, fungi and yeasts. It was shown that bacteria are more effective to accumulate gold than fungi. Nevertheless the mobilization and oxidation of Au by fungi may indeed be significant and thus worthy of consideration in gold biogeochemical cycling as the authors propose. The authors exclude prokaryotes from their experiments but it would be interesting to explore also a batch where prokaryotes are present, in order to determine whether fungi or prokaryotes have largest gold oxidation potential. Similarly, it would be of large interest also to include sequencing of prokaryotes of the anomaly/reference samples, particularly as the fungi sequences showed small differences between the areas. On the same theme, the potential quantitative difference between prokaryotes and fungi for gold mobilization and accumulation should be discussed.*

Response: We agree with Reviewer 1’s suggestion that eukaryotic and prokaryotic microorganisms should be both considered in gold biogeochemical cycling. We believe the interaction between fungi and gold is an indispensable complement to the current view of gold biogeochemical cycling. To facilitate the discussion on the different response between indigenous fungal and bacterial communities to *in situ* gold, we have now sequenced the 16S rRNA genes of the indigenous bacterial community using MiSeq sequencing. The detailed bacterial community structures and diversity indexes were shown in Fig. S4 and Table S5 and S6. The correlation between bacterial diversity and *in situ* gold concentrations has also been

calculated (Table 4). We re-constructed two microcosms to assess the prokaryotic potential on gold oxidation (Line 108-127, Table 1, Fig. 3). We have now discussed the results (Line 311-331).

Fig. S4

Figure S4. Relative abundance of different bacterial phyla present in the gold anomaly (NBD01–05, NBD07–10) and the reference area (NBD11–13, NBD16–18) as estimated by MiSeq sequencing of 16S rRNA genes. Bars indicate the relative abundance of the different phylogenetic groups at phylum level.

Table S5. Bacterial 16S rRNA gene Miseq sequencing parameters and statistical estimators in different sampling sites of the gold anomaly and the reference area.

Sample	No. of sequences	No. of OTUs	Chao	Inverse Simpson's	Shannon
Gold anomaly					
NBD01	1164	767	4110.46	257.64	6.25
NBD02	1164	644	2714.32	153.96	5.92
NBD03	1164	731	3525.40	109.40	6.04
NBD04	1164	765	5583.02	158.22	6.17
NBD05	1164	658	2567.39	75.91	5.75

NBD07	1164	393	1193.28	25.21	4.66
NBD08	1164	606	2444.13	71.29	5.69
NBD09	1164	608	2426.01	80.46	5.72
NBD10	1164	565	2125.71	63.93	5.57
Reference area					
NBD11	1164	648	2812.98	135.28	5.90
NBD12	1164	580	2089.44	110.84	5.71
NBD13	1164	600	2385.03	126.84	5.78
NBD16	1164	534	1839.20	71.96	5.46
NBD17	1164	509	1937.34	81.99	5.42
NBD18	1164	556	2147.34	51.16	5.49

Table S6. The difference of bacterial communities' diversity between the gold anomaly and the reference area

	Gold anomaly vs. Reference		Hotspots vs. Reference	
	Chao	Inverse Simpson's	Chao	Inverse Simpson's
P	0.1821	0.6496	0.1275	0.6403
t, df	t=1.41, df=13	t=0.12, df=13	t=1.73, df=7	t=0.49, df=7
Significant ^a	- ^b	-	-	-

^a, $P < 0.1$ was considered statistically significant.

^b, + and - represented significant and non-significant, respectively.

Line 108-127: We compared the microbial gold oxidation potential in the gold anomaly and the reference area using six batch-type soil microcosms additionally spiked with 40 μM metallic gold particles (Table 1). Microcosm GA6F+cs exhibited two phases of linearly increasing oxidation that were staggered and separate from each other (Fig. 3a). The first phase ranged from the initial time point to 45 h and exhibited a relatively high gold oxidation rate of up to 0.19 $\mu\text{M}/\text{h}$. The second phase occurred from 191 to 453 h, at a lower (0.01 $\mu\text{M}/\text{h}$) oxidation rate and followed substantial Au(III) depletion. The concentration of Au(III) shifted linearly with pH (Pearson $r=0.9936$, $P=0.07$) from 45 to 191 h, a rate change period. The concentration of Au(III) in the reference microcosm RA6F+cs increased only between 0 and 45 h, at a rate of 0.07 $\mu\text{M}/\text{h}$, and remained at *ca.* 5.68 \pm 0.59 μM until the end of the experiment (Fig. 3a). No linear relationship between Au(III) and pH (Pearson $r=-0.5184$, $P=0.65$) was observed from 45 to 191 h.

Intriguingly, the Au(III) indicator 3,3',5,5'-tetramethylbenzidine (TMB, Sigma-Aldrich)²⁵ revealed no conspicuous Au(III) occurrence in microcosm GA8B+cyc (Fig. 3b), with pH stably around 6. The curves for RA8B+cyc and GA8B+cyc were almost identical. In addition, no unusual pH fluctuations were noted in sterile soil microcosms GA6I and RA6I (Fig. 3c). Concurrently, Au(III) concentration in either microcosm underwent only minimal change, with the exception of an increase (up to $8.48 \pm 1.98 \mu\text{M}$) in the first 17 h, attributable to a gradual release of loosely bound gold ions from soil matrices and chemical oxidation (e.g., amino acids-mediated oxidation⁹) following soil sterilization²⁶.

Table 1. Microcosm conditions

Microcosms	Sampling sites		Sampling dates		Biological activity			Antibiotics supplementation	
	Gold anomaly (G)	Reference area (R)	Aug. 2016 (A6)	Aug. 2018 (A8)	Fungi (F)	Bacteria (B)	Inactive (I)	Cycloheximide (+cyc)	Chloramphenicol and streptomycin (+cs)
GA6F+cs ^a	●		●		●				●
GA8B+cyc	●			●		●		●	
GA6I	●		●				●		
RA6F+cs		●	●		●				●
RA8B+cyc		●		●		●		●	
RA6I		●	●				●		

^a, a black dot means the condition was applied to construct the microcosm.

Fig. 3

Figure 3. Microbial gold oxidation potential and geochemical modeling in soil microcosms. (a–c) Time-dependence between Au(III) concentration and pH in different microcosms expressed as means of triplicates \pm standard deviation. (a) Gold oxidation in fungal microcosms GA6F+cs and RA6F+cs, with \square and \square marking the first and second linear phases of gold oxidation in microcosm GA6F+cs. (b) Bacterial microcosms GA8B+cyc and RA8B+cyc. (c) Sterilized microcosms GA6I and RA6I. (d, e) Geochemical models showing predominant Au speciation as a function of pH-Eh at 25 °C. The systems contain initial Au^+ activity of 10^{-6} to 10^{-12} , and HCO_3^- activity of 0.01 (d) or $\text{S}_2\text{O}_3^{2-}$ activity of 0.01 (e). The carbon and sulfur species are reacted with the axis species, and the dashed grey lines show the boundary of predominant carbon and sulfur species over pH-Eh conditions. Predominant Au speciation at 1 part per trillion (ppt) (solid red line), 1 part per billion (ppb) (dashed orange line), and 1 part per million (ppm) (dashed yellow line) is indicated. Blue arrows show the direction of gold mineralization.

Line 311-331: According to the soil microcosm result, indigenous fungi have the potential to markedly influence ionic gold formation in the gold anomaly. In the fungal microcosm GA6F+cs containing the gold anomalous soil spiked with metallic gold (Fig. 3a), the maximum Au(III) concentration occurred at 119 h and corresponded to about 33.6% of the total gold weight in the microcosm, whereas the sterilized control GA6I showed no increase in Au(III) after the initial period (Fig. 3c). The results confirmed that biological redox transformation was the main factor to mediate ionic gold formation in the microcosms. We found that the added metallic gold in microcosms GA6F+cs could not be completely converted to Au(III), indicating that the biological oxidation process was competing with gold reduction and precipitation. Given that the transformation of Au(III) to metallic gold is auto-catalyzed³⁹, fungal growth phase and metabolism could be critical to gold oxidation. Indeed, Au(III) concentration decreased drastically in microcosm GA6F+cs from 453 h and leveled to that in the reference microcosm, indicating that gold oxidation became attenuated whereas Au(III) reduction and accumulation became dominant in the declining phase. Compared with fungal microcosm GA6F+cs, we detected a weak Au(III) signal in the bacterial microcosm GA8B+cyc containing the gold anomalous soils spiked with metallic gold (Fig. 3b), indicating that the indigenous bacterial consortium had either a low gold oxidation potential or a fast Au(III) turnover. Previous studies revealed that bacteria could rapidly reduce and precipitate Au(III)^{40,41}. The rapid reduction and precipitation of Au(III) in GA8B+cyc may hinder colorimetric determination. In fact, a previous study, which did not distinguish between dissolved and colloidal gold, showed that prokaryotic microorganisms could oxidatively dissolve 45.7% (in weight) of gold in soil microcosms after 45 days of incubation⁴².

Comment 19: *Line 321: “The result agrees with a previous finding that native gold could be dissolved by resident microbiota in auriferous soils⁴³”. The experimental setup (Fig 9 and related parts) of the current study seems to be quite similar to reference 43, which used prokaryotes. A bit more detailed comparison with that study would be appropriate.*

Response: The major difference between that aspect of the experimental setup in our paper and the reference is the methods used to detect the gold species in the microcosms. We applied Au(III)-specific probe TMB to colorimetrically determine Au(III) concentration, while the authors of reference 43 used Inductively Coupled Plasma Mass Spectrometry (ICP-MS). ICP-MS does not distinguish between ionic and metallic gold. Since the main objective of the study is to detect ionic gold in the microcosms, ICP-MS is not suitable to be applied in our experiment. We have now added the discussion (Line 323-331).

Line 320-328: Compared with fungal microcosm GA6F+cs, we detected a weak Au(III) signal in the bacterial microcosm GA8B+cyc containing the gold anomalous soils spiked with metallic gold (Fig. 3b), indicating that the indigenous bacterial consortium had either a low gold oxidation potential or a fast Au(III) turnover. Previous studies revealed that bacteria could rapidly reduce and precipitate Au(III)^{40,41}. The rapid reduction and precipitation of Au(III) in GA8B+cyc may hinder colorimetric determination. In fact, a previous study, which did not distinguish between dissolved and colloidal gold, showed that prokaryotic microorganisms could oxidatively dissolve 45.7% (in weight) of gold in soil microcosms after 45 days of incubation⁴².

Comment 20: Line 322: “Furthermore, the Au(III) concentration curve in the microcosms of the reference area was similar to that in the sterilized controls” – there is actually an increase in the beginning and then constant values. The anomaly samples show similar increase (but larger) of oxidation potential. Can this be due to the higher Au concentrations of the anomaly sample or are the concentrations of indigenous gold in the sample negligible compared to the added concentrations? There is a need for more information about the nature of the material used in the experiment. After a while, the Au (III) decreases in the anomaly sample, to levels similar to the reference sample. Is this due to accumulation of gold? This needs to be discussed. There is some discussion about it at line 347, but needs to be expanded (and reference given to the figure the discussion is about).

Response: Many thanks for this comment. We have now discussed the “increase” in Line 345-350 and added the conditions of the microcosms in Table 1. The indigenous gold concentrations of the samples have now been shown in Table S1. The indigenous gold concentrations were negligible compared to the added concentration. It is very likely that gold reduction and accumulation lead to the attenuation of Au(III) concentration because the transformation of Au(III) to metallic gold is auto-catalyzed³. We have now discussed the results (Line 317-323).

Line 345-350: Superoxide is a common microbial metabolic by-product¹⁹. Therefore, superoxide-initiated gold oxidation should be a ubiquitous and non-selective process. Indeed, we obtained Au(III) signals in both fungal microcosms, GA6F+cs and RA6F+cs (Fig. 3a). However, relative to GA6F+cs, only a slight linear increase in Au(III) was detected in RA6F+cs between 0 and 45 h (Fig. 3a), indicating that the fungal consortium in the gold anomaly possessed a high gold-oxidizing capacity compared to fungi present in the non-auriferous environment.

Line 317-323: We found that the added metallic gold in microcosms GA6F+cs could not be completely converted to Au(III), indicating that the biological oxidation process was competing with gold reduction and precipitation. Given that the transformation of Au(III) to metallic gold is auto-catalyzed³⁹, fungal growth phase and metabolism could be critical to gold oxidation. Indeed, Au(III) concentration decreased drastically in microcosm GA6F+cs from 453 h and leveled to that in the reference microcosm, indicating that gold oxidation became attenuated whereas Au(III) reduction and accumulation became dominant in the declining phase.

Comment 21: *Line 353: “In parallel, the redundancy analysis illustrated that the soil carbon content became a less important restricting factor in shaping the fungal community in the gold anomaly than in the reference area.” In general I am not overly enthusiastic about PCA, and I think it should be used with caution, such as in this case and the following interpretations. Particularly since the two areas showed no significant difference in carbon contents. Therefore, in my view, the conceptual model is not convincingly supported by the data.*

Response: We have now rephrased the text and reduced emphasis in the discussion about the importance of CCA analysis results as Reviewer 1 has suggested (Line 379-381). We have also removed the conceptual model accordingly.

Line 379-381: Separately, CCA verified that gold significantly affected the composition of the indigenous fungal community in the gold anomaly (Fig. 8).

Comment 22: *Supplementary tables*

Please write out all abbreviations at first occasion.

Include Au concentrations of the sequencing samples if possible.

Include chemical composition of all soil samples.

Response: We have now ensured all the abbreviations in the main text and SI are in full at the first occasion. The gold concentrations of the sequencing samples are listed in Table S1. The chemical composition of all soil samples is listed in Table S2.

Reviewer #2 (Remarks to the Author):

Comment 23: *The work is related to the geomicrobiological transformations of precious metal gold by fungi and could be of great interest for Nature Communications readers. The manuscript can be published after major revision. The main strengths of this work are the combination of multidisciplinary approaches to geomycological study of fungal communities of the gold anomaly in the Golden Triangle Gold Prospect including culture-independent and culture-dependent methods as well as the authors' finding that gold may alter the fungal growth on carbon sources (e.g. sucrose) under laboratory conditions.*

Response: We thank Reviewer 2 for the encouraging comments.

Comment 24: *However, there are following main problems with the current version of the manuscript:*

(1) In some places the data interpretation in the text is not very clear and sometimes confusing, for example: on page 5, the last paragraph regarding the gold, it's not clear whether it was about Fig.2 and if yes there was no significant difference in gold concentration between the anomaly and the reference; there was heterogeneous distribution of gold in the anomaly with site where gold reached 40 ng/g;

Response: In the last paragraph on page 5 of the former version, we attempted to describe the distribution pattern and mineralogical properties of *in situ* gold in the gold anomaly and the reference area. We agree with Reviewer 2 that gold was heterogeneously distributed and there are gold hotspots in the gold anomaly as also suggested by Reviewer 1. We have now compared the geochemical parameters of the hotspots vs. the reference area with much greater differences between the Au concentrations (Line 90-102, Fig. 2). Considering the high heterogeneity of the soil samples, we decided to use $P < 0.05$ instead of $P < 0.001$ to represent statistical significance of the geochemical parameters. The detailed geochemical data regarding all sampling sites have now been shown in SI (Fig. S1). One tailed t-test showed gold concentrations in the gold anomaly are highly discrete from that in the reference area ($P = 0.08$). Although the P value is > 0.05 , the geological significance of *in situ* gold concentrations between the two areas should not be ignored^{4,5}.

Line 90-102: The surface soil of the Golden Triangle Gold Prospect is mildly acidic and iron-laden (Fig. S1, Table S2). A comparison between the two areas, including all sampling sites, showed no statistically significant differences in physicochemical parameters except for sulfur content (Fig. S1; Table S2). In particular, a one-tailed t -test showed gold concentrations in the gold anomaly were highly discrete from those in the reference area ($P = 0.08$, Fig. S1). Gold was found to be present as < 200 nm particles, possibly within the lattice of minerals in the surface soil and distributed heterogeneously in the anomaly to form hotspots. The discrete localized gold concentrations could be up to 40 ng/g, thus elevating the mean gold concentration in the gold anomaly. Sampling sites NBD02, NBD03, and NBD10 with *in situ* gold concentrations 1.5-fold greater than or equal to the median (3.54 ng/g) were determined as gold hotspots

in the gold anomaly (Fig. 1). Statistically significant differences could be detected between the hotspots and the reference area regarding *in situ* gold concentration; whereas other geochemical parameters (e.g., soil pH, electrical conductivity (Ec), carbon, nitrogen, sulfur, calcium, iron, and water content) were similar (Fig. 2).

Fig. 2

Figure 2. Soil geochemical parameters of the hotspots (sites NBD02, NBD03, and NBD10) and the reference area in the Golden Triangle Gold Prospect. The significance of the differences between two sets of data was determined by an unpaired two-tailed *t*-test. $P < 0.05$ was considered statistically significant. The error bar indicates the standard deviation.

Fig. S1

Figure S1. Soil geochemical parameters of the gold anomaly (10 sampling sites, Table S1) and the adjacent reference area. The box extends from the 25th to the 75th percentile. The line in the middle of the box was plotted at the median. The whiskers go down to the smallest value and up to the largest value. The significance of the differences between two sets of data was determined by an unpaired *t*-test (GraphPad Prism Version 7). In particular, the *P* value for gold content was based on a one-tailed *t*-test calculation. *P*<0.05 was considered statistically significant.

Comment 25: Talking of ESEM it is not clear what “invisible” gold is: did authors mean that gold content in the sample was below the sensitivity threshold of the EDX analysis? But the gold speciation (e.g. association with clay minerals or soil particles) should not affect the registration of gold as the

element in the specimen; the absence of gold in ESEM specimen could be also explained by high heterogeneity of gold distribution;

Response: We agree with Reviewer 2 and have removed the term “invisible gold” as this could be misunderstood. Invisible gold is commonly used by geologists, but not a general readership audience. The standard deviation in the amount of gold (7.65 ± 3.73 ng/g) in the anomalous area would indicate that the gold is heterogeneously concentrated or present in relatively large particles compared the more evenly distributed gold (2.04 ± 0.39 ng/g) in the background areas. A scanning electron microscope (Philips XL40 operating in controlled pressure mode with a chamber pressure of 0.5 mBar and an accelerating voltage of 30 kV) was used to examine samples from the anomalous area. Typically, we would expect particles larger than about 0.5 μm to be relatively easy to locate, however, no gold particles were observed which indicates that the gold may be present as particles smaller than about 200 nm or possibly incorporated into the lattice of associated minerals. There is also the possibility of just a few large nuggets over a micron in size that were not observed. Therefore, we have now revised the paragraph (Line 94-96).
Line 94-96: Gold was found to be present as <200 nm particles, possibly within the lattice of minerals in the surface soil and distributed heterogeneously in the anomaly to form hotspots.

Comment 26: *On pages 6-7, the paragraph with lines 110-126, text regarding the setting of two microcosms is not very clear (here and also on page 20): it could be said simply that there were two kinds of microcosms with (1) reference soil and (2) anomaly soil additionally spiked with gold particles; also to add the speciation (metallic) of gold particles on line 111, the first time of their mentioning instead of line 121;*

Response: In the revised manuscript, we now have six batch-type soil microcosms including two new microcosm experiments (prokaryotic activity in the gold anomaly vs. the reference) to address the comments of the reviewers fully. The conditions of the microcosms are shown in Table 1. In addition,

we have now added more information to the method section (Line 422-448) and the speciation (metallic) of gold was added to the sentence the first time of its mentioning (Line 108-109).

Table 1. Microcosm conditions

Microcosms	Sampling sites		Sampling dates		Biological activity			Antibiotics supplementation	
	Gold anomaly (G)	Reference area (R)	Aug. 2016 (A6)	Aug. 2018 (A8)	Fungi (F)	Bacteria (B)	Inactive (I)	Cycloheximide (+cyc)	Chloramphenicol and streptomycin (+cs)
GA6F+cs ^a	●		●		●				●
GA8B+cyc	●			●		●		●	
GA6I	●		●				●		
RA6F+cs		●	●		●				●
RA8B+cyc		●		●		●		●	
RA6I		●	●				●		

^a, black dot means the condition was applied to construct the microcosm.

Line 422-448: Six batch-type microcosms were set up under aerobic conditions with metallic gold particles (1.5–3.0 μm ; Sigma-Aldrich; St Luis, MO, USA) supplementation. Conditions for all microcosms are listed in Table 1. Briefly, 20 g wet weight of fresh organic-rich soil was added to sterile 250-mL screw cap Erlenmeyer flasks (PYREX®, Sigma-Aldrich) with 200 mL of sterile Milli-Q water. For microcosms GA6F+cs and RA6F+cs, the mixture was supplemented with 0.2- μm filter-sterilized 30 $\mu\text{g}/\text{mL}$ chloramphenicol (Fisher Scientific, Waltham, MA, USA) and 100 $\mu\text{g}/\text{mL}$ streptomycin sulfate (Fisher Scientific) from ethanolic or aqueous stocks, respectively, to inhibit bacterial growth⁵⁴. For microcosms GA8B+cyc and RA8B+cyc, the mixture was supplemented with 5% cycloheximide (0.2- μm filter-sterilized) to inhibit fungal growth²⁹. For microcosms GA6I and RA6I, the mixture was sterilized by repeated autoclaving (121 °C, 20 min, twice over 24 h) to cease biological activities. We added sterilized gold particles to the microcosms to achieve an initial gold concentration of 40 μM . The microcosms were inoculated statically in the dark at 10 °C. Before each sampling, the microcosms were homogenized by gentle shaking. The microcosms GA6F+cs, RA6F+cs, GA6I, and RA6I were sampled at 0, 17, 45, 119, 191, 310, 453, and 693 h; whereas the microcosms GA8B+cyc and RA8B+cyc were sampled at 0, 47.5, 140.5, 185.5, 286.5, and 693 h. We pipetted 1 mL of slurry from each flask to colorimetrically determine the Au(III) concentration using TMB (Sigma-Aldrich) and measured the resulting absorbance at 654 nm²⁷

with a DR 5000™ UV-Vis spectrophotometer (Hach, Dandenong South, VIC, Australia). The slurry was centrifuged at $3000 \times g$ for 1 min and 0.5 mL supernatant was pipetted into a cuvette (BRAND® standard disposable cuvettes, Sigma-Aldrich). Then, 0.1 mL of TMB indicator was added to the cuvette and mixed with the supernatant by gentle pipetting. The mixture was incubated at 25 °C in the dark for 20 min before the colorimetric measurement. We prepared the TMB indicator by dissolving TMB in a mixture of ethanol (100%, Sigma-Aldrich) and 1.0 M sodium acetate/acetic acid buffer (pH 3.5) at a volume ratio of 4:1. The concentration of TMB in the solution was 2.0 mM. The standard curve was prepared using AuCl₃ (Sigma-Aldrich) in Milli-Q water solution at a final concentration of 2, 3, 10, 20, and 30 μM (Table S11, Fig. S6). The pH of the microcosms was also monitored at the sampling time using a pH meter (MC-80; TPS). The microcosm setup and all measurements were conducted in triplicate.

Line 108-109: We compared the microbial gold oxidation potential in the gold anomaly and the reference area using six batch-type soil microcosms additionally spiked with 40 μM metallic gold particles (Table 1).

Comment 27: *On line 124 – are the words about linear relationship related to the correlation analysis pH vs Au(III)? if not it is recommended to check this correlation and to present correlation coefficients and to compare data on current Fig.3.*

Response: We have now specified the correlation between pH and Au(III) occurred in Microcosm GA6F+cs from 45 to 191 h which was a rate change period. The Pearson correlation coefficients have now been calculated. We have added the data in the context (Line 115-117).

Line 115-117: The concentration of Au(III) shifted linearly with pH (Pearson $r=0.9936$, $P=0.07$) from 45 to 191 h, a rate change period.

Comment 28: *also, somewhere in the paper, it will be very useful to add a COMPREHENSIVE DIAGRAM OF BIOGEOCHEMICAL TRANSFORMATIONS OF GOLD including redox processes, changes in gold speciation, known minerals etc. which will clarify this work, upgrade it and improve its perception by the diverse readers of Nature Communications in general (and will make Fig.3 better understood);*

Response: Many thanks for this constructive comment. As we understood, the “COMPREHENSIVE DIAGRAM OF BIOGEOCHEMICAL TRANSFORMATIONS OF GOLD” could be a thermodynamic model to depict the gold redox process. Therefore, we have now constructed two geochemical models (carbon- and sulfur-rich, respectively) (Line 140-155, Fig 3d, e). In the models, we showed the redox processes and changes in gold speciation. We further revealed reactive oxygen species (e.g., superoxide) is a more feasible oxidant relative to molecular oxygen through the thermodynamic calculation. Following the geochemical model, we have also drawn a predictive model to explain the potential mechanism of mycological gold redox transformation and added it to SI (Fig. S5).

Line 140-155: We proposed two geochemical models to decipher gold redox transformation in the soil microcosms under carbon- and sulfur-rich conditions, respectively. Gold species composition and solubility varied widely depending on pH, redox potential (Eh), and the presence of carbon and sulfur species. In the carbon-rich system (Fig. 3d), an increase in Eh causes the predominant carbon species to change from methane (CH₄) to carbon and then to carbonate/bicarbonate. Under oxidizing conditions, gold is transported as hydrated Au³⁺ around pH 0–1 or hydroxyl complex Au(OH)₄⁻ around pH 1–14. The pH-Eh boundary of the O₂⁻/H₂O redox couple is close to the boundary of Au(OH)₄⁻/Au(OH)₂⁻. With a decrease in Eh, Au³⁺ is reduced to Au⁺ with the predominant species becoming Au(OH)_(aq) and Au(OH)₂⁻. Under reducing conditions, the predominant Au species is Au⁰_(s), which is the main mineral form of gold. Organic matter is likely the major chelator in gold mineralization. Under sulfur-rich conditions (Fig. 3e), thiosulfate anion (S₂O₃²⁻) is not stable in acidic conditions due to disproportionation. The predominant sulfur species is sulfate (HSO₄⁻ and SO₄²⁻) under oxidizing conditions and sulfide (H₂S and HS⁻) under

reducing conditions. Gold is transported as hydroxyl complexes under oxidizing conditions and Au-HS complexes under reducing conditions. Thus, in both carbon- and sulfur-rich systems, the formation of soluble gold species is affected by the concentration of gold, as well as pH and Eh.

Fig. 3

Figure 3. Microbial gold oxidation potential and geochemical modeling in soil microcosms. (a–c) Time-dependence between Au(III) concentration and pH in different microcosms expressed as means of triplicates \pm standard deviation. (a) Gold oxidation in fungal microcosms GA6F+cs and RA6F+cs, with ① and ② marking the first and second linear phases of gold oxidation in microcosm GA6F+cs. (b) Bacterial microcosms GA8B+cyc and RA8B+cyc. (c) Sterilized microcosms GA6I and RA6I. (d, e)

Geochemical models showing predominant Au speciation as a function of pH-Eh at 25 °C. The systems contain initial Au^+ activity of 10^{-6} to 10^{-12} , and HCO_3^- activity of 0.01 (d) or $\text{S}_2\text{O}_3^{2-}$ activity of 0.01 (e). The carbon and sulfur species are reacted with the axis species, and the dashed grey lines show the boundary of predominant carbon and sulfur species over pH-Eh conditions. Predominant Au speciation at 1 part per trillion (ppt) (solid red line), 1 part per billion (ppb) (dashed orange line), and 1 part per million (ppm) (dashed yellow line) is indicated. Blue arrows show the direction of gold mineralization.

Fig. 5S

Figure S5. The predictive model for mycological gold redox transformation under Earth surface conditions. Dashed arrows indicate superoxides and ligands are produced intracellularly.

Comment 29: On page 7, line 131, what are the reasons for choosing autoclaving of gold particles over other ways of sterilization (oven-drying, treatment with absolute alcohol etc.) if it could trigger abiotic oxidation, what is the extend of this oxidation, have the authors study it (qualified (changes in speciation) and quantified (how much)) to claim that the alterations of metallic gold was negligible?

Response: Pure metallic gold is not affected by autoclaving⁶. The chemical oxidation we mentioned here is a process between gold and soil chemicals following the sterilization performance which may change soil chemical properties. We answered a similar question in Response 46: Many methods have been applied for soil sterilization⁷. Soil microorganisms are effectively eliminated by autoclaving, gamma-irradiation, propylene oxide, and mercuric chloride⁸. These methods can be approximately grouped into two categories: physical and chemical methods. To avoid the influence of the external substances on gold oxidation, we excluded all the chemical methods in our study such as propylene oxide and mercuric chloride. The physical methods, autoclaving and gamma-irradiation, both can alter soil composition and structure⁹. We constructed a similar microcosm with gamma-irradiation (24 h, ChemCentre, Western Australia) treated soils. Au(III) slightly increased from 0 to 72 h, which is similar to the pattern of the soil microcosm sterilized by autoclaving. We believe the Au(III) signals were induced by the reaction between the added metallic gold particles and soil chemicals (e.g., amino acids⁷) following the treatment¹⁰. In addition, although gamma-irradiation may induce less organic matter alteration, iron oxide minerals can be significantly reduced¹¹. Iron oxides are the dominant minerals in our system. Since Fe²⁺/Fe³⁺ redox couple stimulated Fenton reaction can affect gold speciation¹², we did not use gamma-irradiation to sterilize the ferruginous soils in the experiment. In addition, pasteurization is common in agricultural application to kill pathogenic organisms and seeds but is not an efficient method relative to the aforementioned methods to eliminate microorganisms in a long-term soil incubation experiment¹³. Therefore, we believe repeated-autoclaving, although is of limitations, is the most appropriate method to be applied in our study.

Comment 30: *On page 11, lines 204-205, it should be “physicochemical”, not “physiochemical”;*

Response: Fixed.

Comment 31: *On page 12, lines 217-222, to clarify it better (simpler) in the text and to specify parameters (line 220)*

Response: We have now revised the paragraph and reanalyzed the pattern of the environmental variables (Line 268-271).

Line 268-271: Canonical correspondence analysis (CCA) further demonstrated that *in situ* gold content was an important factor explaining fungal community composition in the gold anomaly, but not in the reference area (Fig. 8). Moreover, *in situ* gold content showed a strong negative correlation with soil pH and intimate association with Hypocreales in the gold anomaly.

Comment 32: *On page 12, line 227, what are the reasons of choosing 18S as a phylogenetic marker over universally used ITS?*

Response: In the first version of our manuscript, the 18S rRNA gene primer set EF4/EF3¹⁴ was used for fungal ribosome RNA gene amplification. The primer set can amplify Ascomycota, Basidiomycota, and Zygomycota. We attempted to amplify the DNA of unknown fungal isolates from the environment by first using this primer set. We agree with Reviewer 2 that ITS is used more often and can be more precise. Thus, we have now done the phylogenetic analysis using ITS as the marker. According to the sequenced ITS, we assigned the phylogeny of TA-pink1 to *Fusarium oxysporum* (Line 159-161).

Line 159-161: The sequenced internal transcribed spacer (ITS) region of TA_pink1 was identical (100%) to that of the model strain *Fusarium oxysporum* CP13, which belongs to the order Hypocreales and has not been reported as a gold-oxidizing fungus.

Comment 33: *(2) About fungal growth*

on page 15-16, strictly speaking, it was all about the extension of fungal colony on the surface of agar medium; extension is an indicator of colonization but, in many cases, it does not reflect the changes in

biomass because the morphology of fungal mycelium can be altered by different growth conditions and a fungus can produce either very thick dense mycelium (phalanx strategy) or, in contrast, long sparsely branched, branchless hyphae (guerrilla strategy); did authors take into account the morphology of fungal mycelium? in case of this paper, the colonization of environment is very important for fungi competing for resources, but it should be specified that it is only an extension (surface growth)

Response: This is a valuable comment. We agree that our results are about fungal extension. The morphology of the fungal mycelium was not considered in the manuscript. We have now rephrased the context to express this point clearly (Line 212-213).

Line 212-213: Hyphal extension analysis was carried out to explore the ecophysiological influence of gold on TA_pink1 (Fig. 6; Table 3).

Comment 34: *on page 16, Fig. 9, the fungal colony extension is linear, however often there is a delay of surface (or any) growth which is called lag-phase indicating that the microbial culture needs some adaptation to the new environment (the more favorable environment the less the duration of lag-phase); it is obvious that the fungus TA pink 1 manifested the lag-phase (extension delay) on sucrose medium on Fig. 9a and the extension rate of linear fungal surface growth on sucrose should be calculated from 284h up to around 500h, and similarly the extension rate calculations of the other data should take into account the lag-phase (growth delay) and be performed ONLY IN THE LINEAR AREA OF THE PLOT (in current version of the manuscript, there was no point to calculate near-lag-phase areas data and compare them, plus there was no any statistics); I would strongly recommend the authors to recalculate the extension rates and to make statistical analysis whether there is a significant difference between extension rate of the linear growth on Fig. 9 data as well as to identify the lag-phases durations by the point of cross of the straight line of linear fungal growth with horizontal line parallel to the time axis [see the diagram enclosed, also see Principles of Microbe and Cell Cultivation by S.J. Pirt]; so far, considering that fungus TA pink 1 was struggling to grow on sucrose under control conditions compared*

to lignin and looking at the Fig. 9 plots, the main finding is that gold can alter the surface growth of the fungus by reducing its adaptation period (lag-phase duration) to the alternative carbon source sucrose resulting in speeding up the colonization of the environment.

Response: Many thanks for the valuable comments. We have now calculated the lag-phases of the curves and labeled them on Fig. 6 (former Fig. 9) according to the reference the reviewer suggested. The extension rates have also been calculated in the linear area of the plot with linear regression analysis (Line 212-223, Table 3). We can confirm that there is no statistical difference between extension rates of the linear growth with and without gold supplement. However, the gold supplement markedly reduced the lag-phase when the fungus grew on sucrose and attenuated the colony diameter variation, whereas gold slightly reduced the lag-phase from 83.0 h to 79.9 h when lignin was used as carbon source.

Line 212-223: Hyphal extension analysis was carried out to explore the ecophysiological influence of gold on TA_pink1 (Fig. 6; Table 3). Sucrose and lignin were chosen as the sole carbon sources in the growth medium because they are common carbohydrate species in plant root exudates²⁹ and residues³⁰. The lag-phase was calculated according to its quantitative definition³¹. TA_pink1 showed a long lag-phase (195.7 h) and drastic colony diameter variation when grown on Czapek Dox agar³² with sucrose as carbon source (Fig. 6a; Table 3). Interestingly, gold supplementation reduced the lag-phase to 38.6 h and attenuated the variation in colony diameter, even though linear extension rates were not altered. By contrast, gold exhibited different effects on hyphal extension when lignin was used as carbon source (Fig. 6b; Table 3). Without gold supplementation, the slope of the linear extension period was 0.081 ± 0.001 mm/h and the lag-phase was 83.0 h. Gold slightly accelerated the extension rate (0.088 ± 0.0005 mm/h) and reduced the lag-phase (79.9 h), although the patterns of the two extension curves were nearly identical.

Fig. 6

Figure 6. Hyphal extension of TA_pink1 with (a) sucrose and (b) lignin as the sole carbon source. Error bars denote the standard deviation of the mean of triplicate measurements and are only shown when greater than the symbol dimension.

Table 3. Linear regression analysis of hyphal extension

	Sucrose		Lignin	
	No gold	With gold	No gold	With gold
Best-fit values ± SE				
Slope	0.070±0.005	0.078±0.0005	0.081±0.001	0.088±0.0005
Y-intercept	-12.38±1.9	-2.05±0.2	-5.45±0.5	-6.73±0.2
X-intercept	177.90	26.36	66.96	76.67
Goodness of Fit				
R square	0.9947	0.9999	0.9990	0.9999
Equation	Y = 0.07×X - 12.4	Y = 0.078×X - 2.1	Y = 0.081×X - 5.5	Y = 0.088×X - 6.7
Differences of the linear extension rates				
P value		0.1		0.01
Significance (P < 0.01)		NO		NO
Lag-phase (h)	0-195.7	0-38.6	0-83.0	0-79.9

Comment 35: (3) *And again, as the title of the manuscript is “Evidence for geomycological gold redox transformation under supergene condition” and the manuscript would greatly benefit from the reinforcing the authors’ hypothesis with so called visual summary - a diagram/model and I would strongly recommend making a COMPREHENSIVE DIAGRAM OF BIOGEOCHEMICAL TRANSFORMATIONS OF GOLD BY FUNGI*

Response: Please see Response 28.

Reviewer #3 (Remarks to the Author):

Comment 36: *The manuscript by Bohu and colleges used a combination of molecular, culture-based, and geochemical techniques to identify fungi that can potentially oxidize gold in near surface environments. Using microcosms, they found that gold oxidation occurs more readily than abiotic controls. Furthermore, they were able to obtain a fungal isolate that is capable of oxidizing gold. This is intriguing because gold is typically considered to be biologically inactive. While this manuscript has merit and I see no technical flaws, there are two issues that once fixed would make this a publishable manuscript.*

Response: We appreciate Reviewer 3 for the encouraging comments. Please see the following response to the concerns.

Comment 37: *First, the discussion over extends the interpretation of the results. None of the methods that were employed are able to deconvolve the mechanism(s) of gold oxidation by this fungus or the ecological interpretation with carbon substrates. Some of this can be in the discussion, but it needs to be toned down and should not be the bulk of the discussion. The discussion should mirror the introduction.*

Response: Many thanks for the constructive comments. We have rephrased and toned down the discussion extensively (Line 307-397).

Line 307-397: Dissolved gold species (e.g., Au(I/III) complexes) and colloidal gold are commonly found in the Critical Zone^{5,37}, and are considered the source of most secondary gold deposits². Given that abiotic formation of ionic gold requires high ionizing energy under Earth surface conditions³⁸, the interaction between gold and soil biota may be a critical driver of the redox process.

According to the soil microcosm result, indigenous fungi have the potential to markedly influence ionic gold formation in the gold anomaly. In the fungal microcosm GA6F+cs containing the gold anomalous soil spiked with metallic gold (Fig. 3a), the maximum Au(III) concentration occurred at 119 h and corresponded to about 33.6% of the total gold weight in the microcosm, whereas the sterilized control GA6I showed no increase in Au(III) after the initial period (Fig. 3c). The results confirmed that biological redox transformation was the main factor to mediate ionic gold formation in the microcosms. We found that the added metallic gold in microcosms GA6F+cs could not be completely converted to Au(III), indicating that the biological oxidation process was competing with gold reduction and precipitation. Given that the transformation of Au(III) to metallic gold is auto-catalyzed³⁹, fungal growth phase and metabolism could be critical to gold oxidation. Indeed, Au(III) concentration decreased drastically in microcosm GA6F+cs from 453 h and leveled to that in the reference microcosm, indicating that gold oxidation became attenuated whereas Au(III) reduction and accumulation became dominant in the declining phase. Compared with fungal microcosm GA6F+cs, we detected a weak Au(III) signal in the bacterial microcosm GA8B+cyc containing the gold anomalous soils spiked with metallic gold (Fig. 3b), indicating that the indigenous bacterial consortium had either a low gold oxidation potential or a fast Au(III) turnover. Previous studies revealed that bacteria could rapidly reduce and precipitate Au(III)^{40,41}. The rapid reduction and precipitation of Au(III) in GA8B+cyc may hinder colorimetric determination. In fact, a previous study, which did not distinguish between dissolved and colloidal gold, showed that

prokaryotic microorganisms could oxidatively dissolve 45.7% (in weight) of gold in soil microcosms after 45 days of incubation⁴².

The oxidative dissolution of gold mediated by microorganisms, mainly thiosulfate- and cyanide-producing bacteria³, is assumed to use molecular oxygen (O₂) as the electron acceptor^{43,44}. However, the geochemical models revealed superoxide (O₂⁻) rather than molecular oxygen was a more feasible oxidant for the generation of Au(III) in carbon- and sulfur-rich environments, through the following reactions (Fig. 3d, e; see supplementary material for more details):

The pH-Eh boundary of the O₂⁻/H₂O redox couple is close to the boundary of the Au(OH)₄⁻/Au(OH)₂⁻ couple, further supporting the role of superoxide in gold oxidation under Earth surface conditions (Fig. 3d, e). According to reaction (3), gold oxidation is prone to happen at low pH, which is consistent with the correlation of pH and Au(III) concentrations in the rate change period of microcosm GA6F+cs (Fig. 3a).

Superoxide is a common microbial metabolic by-product¹⁹. Therefore, superoxide-initiated gold oxidation should be a ubiquitous and non-selective process. Indeed, we obtained Au(III) signals in both fungal microcosms, GA6F+cs and RA6F+cs (Fig. 3a). However, relative to GA6F+cs, only a slight linear increase in Au(III) was detected in RA6F+cs between 0 and 45 h (Fig. 3a), indicating that the fungal consortium in the gold anomaly possessed a high gold-oxidizing capacity compared to fungi present in the non-auriferous environment. Considering that high-affinity ligands are required in gold oxidation, it is possible that the indigenous fungi of the gold anomaly secreted gold-specific ligands to stabilize ionic gold in the system. The ligands are likely to be organic molecules rather than thiosulfate anion (S₂O₃²⁻) as suggested by the geochemical models (Fig. 3d, e). LA-ICP-MS and XPS analysis consistently confirmed

that *F. oxysporum* isolate TA_pink1 could mobilize gold and excrete carbonyl compounds during the oxidative dissolution of gold (Fig. 4, Table 2), in agreement with a previous finding showing that quinones, a group of carbonyl-rich compounds, were the key molecules in the oxidative transfer of gold⁴⁵. TA_pink1 is thus likely to be an aurophilic microorganism that influences gold speciation via the exudation of a carbonyl-rich ligand with high affinity for ionic gold. In fact, *Fusarium* has been reported to drive stainless steel biofouling by producing a ligand in microaerobic and oligotrophic degassing systems of a heavy water (D₂O) treatment plant⁴⁶. In addition, the anodic reaction between TA_pink1 and colloidal gold demonstrated that TA_pink1 could initiate electron transfer via colloidal gold in the liquid phase (Fig. 5a). This process may influence nanometer gold particles formation (Fig. 5b, c) and facilitate the oxidative dissolution of gold.

Dissimilar to prokaryotic gold redox transformation, the interaction between fungi and gold is unlikely to involve a detoxification process. Fungal polarized growth under nutrient starvation can induce drastic variations in colony diameter⁴⁷. Au(III) attenuated fungal colony diameter variation and reduced the lag-phase during hyphal extension when sucrose, as opposed to lignin, was the sole carbon source (Fig. 6, Table 3). This finding suggests that TA_pink1 prefers to use sucrose rather than lignin when ionic gold is present to accelerate colonization of the environment. Amenabar and co-workers⁴⁸ revealed that microbial substrate preference was dictated by differences in the energy required to metabolize the substrate rather than the energy recovered from it. Therefore, it seems that ionic gold can reduce the energy demand for fungi to utilize unfavorable substances.

At community level, fungal diversity within the gold anomaly (the hotspots especially) correlated positively with *in situ* gold concentrations (Table 4), suggesting that the two variables had a tendency to increase together rather than through toxicity-driven selection⁴⁹. This was not the case for the reference fungal community and bacterial community in either the gold anomaly or in the reference area. Previous studies have indicated that diversity plays a critical role in biomass production⁵⁰ and ecosystem stability⁵¹. Gold, therefore, is very likely a crucial abiotic factor rather than an inactive element in sustaining fungal

ecosystems in the auriferous environment. Separately, CCA verified that gold significantly affected the composition of the indigenous fungal community in the gold anomaly (Fig. 8). MENs analysis further revealed that fungal interspecific interactions in the gold anomaly and the reference area differed under variable gold concentrations (Fig. 9). Hypocreales (the order of the gold-oxidizing fungus) exhibited the highest centrality in the gold anomaly and intimately associated with other dominant OTUs (Fig. 9a), suggesting that mycological gold redox transformation is likely an important biogeochemical process influencing the indigenous microbiome in the auriferous environment. As a consequence, the redox interaction between fungi and gold can be expected to influence gold distribution in the auriferous environment.

In summary, our study shows that fungi, a major component of the soil microbiome, can mediate gold oxidation under Earth surface conditions. The existence of the gold-oxidizing fungus TA_pink1 suggests fungi are able to substantially impact gold biogeochemical cycling. Fungal metabolites, such as superoxide and carbonyl-rich ligands, plausibly play a role in mycological gold redox transformation (Fig. S5). Unlike in a detoxification-oriented interaction, gold may reduce the energetic constrain for the gold-oxidizing fungus to utilize unfavorable substances and speed up colonization of the surrounding environment. The remarkable centrality of gold-oxidizing fungi underscores mycological gold redox transformation is a critical biogeochemical process for the indigenous microbiome in auriferous environments, which in turn, may profoundly influence gold mobilization and accumulation in the terrestrial ecosystem.

Comment 38: *Second, readability can be improved. Especially for Nature Communications with a broad readership. Most of my specific comments address the readability issue. I also suggest reducing some of the jargon to help reach a broader audience. For example, supergene means different things if you are a geologist versus a molecular biologist.*

Response: Many thanks for the suggestion. The table below lists the jargons that have now been replaced.

Jargon	Term instead¹⁵
supergene	Earth surface
calcrete	carbonate-rich accumulations
bauxitic	aluminum-rich
lateritic residuum	iron-rich gravel
organosol	organic-rich soil
ferruginous duricrust	iron crust

Comment 39: *Line 30: split into two sentences.*

Response: Done (Line 30-34).

Line 30-34: Under Earth surface conditions, gold exhibits enigmatic patterns of transformation and translocation^{2,3}. Various forms of gold⁴, including secondary aggregates, crystalline grains, colloidal nanoparticles, and Au(I/III) complexes are ubiquitous in diverse environments, such as hypersaline waters⁵, iron-bearing nodules⁶, and carbonate-rich accumulations⁷.

Comment 40: *Line 32: What is meant by biological transfer?*

Response: We have now changed “biological transfer” to “bioturbation” to better clarify our meaning (Line 35-36). According to Erik Kristensen *et al.* 2012¹⁶, the meaning of faunal bioturbation in aquatic environments includes all transport processes carried out by animals that directly or indirectly affect sediment matrices. They also proposed that “the effects of bioturbation on other organisms and associated processes (e.g. microbial driven biogeochemical transformations) are considered within the conceptual framework of ecosystem engineering.” The process of bioturbation for metal distribution was also partially discussed in Anand *et al.*, 2016¹⁷.

Line 35-36: Capillary migration, gaseous transport, and bioturbation are the principal mechanisms hypothetically affect gold speciation, complexation, and mobility in the Critical Zone¹.

Comment 41: *Line 34: Awkward sentence that should be removed.*

Response: We deleted the sentence.

Comment 42: *Line 39: change “of” to “on”.*

Response: We have now changed ‘of’ to ‘on’ (Line 42).

Line 42: ..., a dominant bacterial species in biofilms on natural gold nuggets, ...

Comment 43: *Lines 91-94: split into two sentences.*

Response: We have now rephrased the section (Line 90-102).

Line 90-102: The surface soil of the Golden Triangle Gold Prospect is mildly acidic and iron-laden (Fig. S1, Table S2). A comparison between the two areas, including all sampling sites, showed no statistically significant differences in physicochemical parameters except for sulfur content (Fig. S1; Table S2). In particular, a one-tailed *t*-test showed gold concentrations in the gold anomaly were highly discrete from those in the reference area ($P=0.08$, Fig. S1). Gold was found to be present as <200 nm particles, possibly within the lattice of minerals in the surface soil and distributed heterogeneously in the anomaly to form hotspots. The discrete localized gold concentrations could be up to 40 ng/g, thus elevating the mean gold concentration in the gold anomaly. Sampling sites NBD02, NBD03, and NBD10 with *in situ* gold concentrations 1.5-fold greater than or equal to the median (3.54 ng/g) were determined as gold hotspots in the gold anomaly (Fig. 1). Statistically significant differences could be detected between the hotspots and the reference area regarding *in situ* gold concentration; whereas other geochemical parameters (e.g.,

soil pH, electrical conductivity (Ec), carbon, nitrogen, sulfur, calcium, iron, and water content) were similar (Fig. 2).

Comment 44: *Lines 94-98: This exact same information is repeated in figure 2. I suggest stating the actual range of concentrations instead of the statistical parameters.*

Response: Done. We have now rephrased the section (Line 90-102).

Comment 45: *Line 98: Is this percent water content?*

Response: Yes. We have now changed “water” to “water content” (Line 100-102).

Line 100-102: *whereas other geochemical parameters (e.g., soil pH, electrical conductivity (Ec), carbon, nitrogen, sulfur, calcium, iron, and water content) were similar (Fig. 2).*

Comment 46: *Line 111: Autoclaving was used to sterilize both for the soils and gold. This is typically not the best method because minerology can change, carbon can be lost, and often times it is ineffective at sterilization of soil (although spreading the soil out on a large pan seems to help). Some explanation as to why autoclaving was chosen over something like pasteurization and any evidence that it did not affect the outcome of the results would be beneficial.*

Response: Many methods are applied for soil sterilization⁷. Soil microorganisms are effectively eliminated by autoclaving, gamma-irradiation, propylene oxide, and mercuric chloride⁸. These methods can be approximately grouped into two categories: physical and chemical methods. To avoid the influence of the external substances on gold oxidation, we excluded all the chemical methods in our study such as propylene oxide and mercuric chloride. The physical methods, autoclaving and gamma-irradiation, both can alter soil composition and structure⁹. We constructed a similar microcosm with gamma-

irradiation (24 h, ChemCentre, Western Australia) treated soils. A Au(III) peak occurred from 0 to 72 h, which is similar to the pattern of the soil microcosm sterilized by autoclaving. We believe the Au(III) signal were induced by the reaction between the added metallic gold particles and soil chemicals (e.g., amino acids⁷) following the treatment¹⁰. In addition, although gamma-irradiation may induce less organic matter alteration, iron oxide minerals can be significantly reduced¹¹. Iron oxides are the dominant minerals in our system. Since $\text{Fe}^{2+}/\text{Fe}^{3+}$ redox couple stimulated Fenton reaction can affect gold speciation¹², we doubtfully use gamma-irradiation to sterilize the ferruginous soils. Furthermore, autoclaving was used to sterilize soils in a similar microcosm research¹⁸. The authors found no discernable changes to the fractionation of Au were detected in the autoclaved compared to the field-fresh materials. In addition, pasteurization is common in agricultural application to kill pathogenic organisms and seeds but is not an efficient method relative to the aforementioned methods to eliminate microorganisms in soils¹³. Therefore, we believe repeated-autoclaving is the most appropriate method in our study.

Comment 47: *Line 123: move “may establish” after “dynamic equilibrium”*

Response: As pointed out by other reviewers, this sentence belongs in the “discussion” rather than a “result”. Thus, we have now removed the corresponding context to discussion and revised the sentences (Line 317-323).

Line 317-323: We found that the added metallic gold in microcosms GA6F+cs could not be completely converted to Au(III), indicating that the biological oxidation process was competing with gold reduction and precipitation. Given that the transformation of Au(III) to metallic gold is auto-catalyzed³⁹, fungal growth phase and metabolism could be critical to gold oxidation. Indeed, Au(III) concentration decreased drastically in microcosm GA6F+cs from 453 h and leveled to that in the reference microcosm, indicating that gold oxidation became attenuated whereas Au(III) reduction and accumulation became dominant in

the declining phase.

Comment 48: *Line 127: pH is not a great indicator of microbial activity and does not confirm it was sterile.*

Response: Agreed. We have now revised the sentence (Line 122-123).

Line 122-123: In addition, no unusual pH fluctuations were noted in sterile soil microcosms GA6I and RA6I (Fig. 3c).

Comment 49: *Lines 151-155: Use standard naming conventions and stick to one taxonomic level when discussing these organisms. The underscores should not be used.*

Response: We revised the context as suggested (Line 238-239).

Line 238-239: These included Chaetothyriales, Agaricales, Eurotiales, Russulales, Helotiales, Pleosporales, Atheliales, and Capnodiales (Fig. 7a).

Comment 50: *Line 229: Since TA_pink1 has 100% identity, is this really a new strain? Conventional methods to fully name a new strain has not be completed. Until then I would suggest referring to your isolate as “an isolate of F. verticillioides”.*

Response: Agreed. We replaced “strain” to “isolate” in the manuscript.

Comment 51: *Figure 7: Please check the figure contrast for color blind people. Some have difficulty discerning pinks from purples.*

Response: Thank you for showing consideration here. We have now modified the line color in Fig. 4c (former Fig. 7c). The line color series applied in other panels were based on the color database in ColorBrewer.org which is colorblind safe.

Fig. 4

Figure 4. Gold-oxidizing capacity of TA_pink1. (a) TA_pink1 was inoculated at the center of PYG agar plates supplemented with 400 μ M colloidal gold. After 14 days of incubation at 10 $^{\circ}$ C in the dark, a gold-dissolving halo appeared around the central colony, dividing the agar into three zones: center, oxidized zone, and undisturbed zone. (b) Quantification of gold concentrations in different zones of the PYG agar by LA-ICP-MS. The horizontal axis represents the gold signal in counts/s. Error bars represent standard deviation, * P <0.001. (c) Expanded gold profile from XPS analysis. The inset shows profiles of major elements, including signals from O 1s (531.0 eV), C 1s (285.1 eV), N 1s (399.5 eV), and Au 4f (84.0 eV). High-resolution scans of C 1s (d-f) and Au 4f spectra (g-i) from different zones. Gray dots represent the intensity of the spectra in counts per second (cps).

Comment 52: Line 264: Is the single asterisks different from the double asterisks?

Response: Fixed. Both *P* values for * and ** were less than 0.001. Therefore, we changed ** to * (Fig. 4 (former Fig. 7)).

Comment 53: *Line 278: Is it possible that the gold was not reprecipitated, but the imaged gold is what is left from the original colloidal gold that has not dissolved? The difference in minerology has not been determined.*

Response: We added gold(I) chloride into the sterilized PYG liquid media. According to gold chemistry^{3,19}, gold(I) chloride thermodynamically transforms to the complex of colloidal gold and organic matters. The formation of the colloidal gold in the PYG liquid media has also been verified by XANES (data are available from the corresponding author upon request). We investigated the gold complex in the liquid media without TA_pink1 using a scanning electron microscope (Philips XL40), as shown in Fig. S3. Sterilized cotton fiber was put into the mixture to offer a non-biological but similar surface for gold precipitation. The size of the precipitated gold was much larger than what we observed on the surface of the fungal hyphae. Thus, it is very likely the fungus in our study is playing a role to affect the precipitation of nanometer gold particles on the hyphae, although more studies are needed to estimate the effect of chemical precipitation (Line 201-203).

Fig. S3

Figure S3. SEM of gold complexes on the surface of a sterilized cotton fiber.

Line 201-203: Moreover, nanometer gold particles occurred on the hyphae of TA_pink1 (Fig. 5b, c) cultured for 14 days in liquid PYG medium with colloidal gold, whereas much larger gold-organic matter complexes were observed in the sterilized control (Fig. S3).

Comment 54: Lines 295 and 299: *I suggest providing overall growth rates first and then a brief description of the change that occurs at this time period. This is also something that could be expanded upon in the discussion. Why is there a change in the slope at this time period?*

Response: We have now calculated the lag-phases of the curves and labeled them on Fig. 6 (former Fig. 9). The extension rates have also been calculated in the linear area of the plot with linear regression analysis (Line 212-223, Table 3). According to the linear regression analysis, the differences of the linear

extension rates (slope) on sucrose was not statistically significant. The lag-phase, on the contrary, drastically reduced from 195.7 h to 38.6 h. The results have now been discussed (Line 364-372).

Line 212-223: Hyphal extension analysis was carried out to explore the ecophysiological influence of gold on TA_pink1 (Fig. 6; Table 3). Sucrose and lignin were chosen as the sole carbon sources in the growth medium because they are common carbohydrate species in plant root exudates²⁹ and residues³⁰. The lag-phase was calculated according to its quantitative definition³¹. TA_pink1 showed a long lag-phase (195.7 h) and drastic colony diameter variation when grown on Czapek Dox agar³² with sucrose as carbon source (Fig. 6a; Table 3). Interestingly, gold supplementation reduced the lag-phase to 38.6 h and attenuated the variation in colony diameter, even though linear extension rates were not altered. By contrast, gold exhibited different effects on hyphal extension when lignin was used as carbon source (Fig. 6b; Table 3). Without gold supplementation, the slope of the linear extension period was 0.081 ± 0.001 mm/h and the lag-phase was 83.0 h. Gold slightly accelerated the extension rate (0.088 ± 0.0005 mm/h) and reduced the lag-phase (79.9 h), although the patterns of the two extension curves were nearly identical.

Fig. 6

Figure 6. Hyphal extension of TA_pink1 with (a) sucrose and (b) lignin as the sole carbon source. Error bars denote the standard deviation of the mean of triplicate measurements and are only shown when greater than the symbol dimension.

Table 3. Linear regression analysis of hyphal extension

	Sucrose		Lignin	
	No gold	With gold	No gold	With gold
Best-fit values ± SE				
Slope	0.070±0.005	0.078±0.0005	0.081±0.001	0.088±0.0005
Y-intercept	-12.38±1.9	-2.05±0.2	-5.45±0.5	-6.73±0.2
X-intercept	177.90	26.36	66.96	76.67
Goodness of Fit				
R square	0.9947	0.9999	0.9990	0.9999
Equation	Y = 0.07×X - 12.4	Y = 0.078×X - 2.1	Y = 0.081×X - 5.5	Y = 0.088×X - 6.7
Differences of the linear extension rates				
P value		0.10		0.0062
Significance (P < 0.01)		NO		YES
Lag-phase (h)	0-195.7	0-38.6	0-83.0	0-79.9

Line 364-372: Dissimilar to prokaryotic gold redox transformation, the interaction between fungi and gold is unlikely to involve a detoxification process. Fungal polarized growth under nutrient starvation can

induce drastic variations in colony diameter⁴⁷. Au(III) attenuated fungal colony diameter variation and reduced the lag-phase during hyphal extension when sucrose, as opposed to lignin, was the sole carbon source (Fig. 6, Table 3). This finding suggests that TA_pink1 prefers to use sucrose rather than lignin when ionic gold is present to accelerate colonization of the environment. Amenabar and co-workers⁴⁸ revealed that microbial substrate preference was dictated by differences in the energy required to metabolize the substrate rather than the energy recovered from it. Therefore, it seems that ionic gold can reduce the energy demand for fungi to utilize unfavorable substances.

Comment 55: *Line 318: With these methods it is not possible to say that fungi were dominant. This was not measured.*

Response: We agree with Review 3 that the dominance of fungi in the microcosms was not measured. Soil microorganisms can be approximately classified as prokaryotes (bacteria and actinomycetes), fungi, algae and protozoa. Generally, fungi is the secondary abundant microorganisms in soils²⁰. It is routinely accepted that broad-spectrum antibiotic supplement (30 µg/mL chloramphenicol and 100 µg/mL streptomycin sulfate in this study) can effectively kill or inhibit the majority of prokaryotes²¹, making fungi the most abundant microorganism in the microcosms. This method has been applied in fungal incubation studies^{22,23}. We have now rephrased the related discussion section. This sentence was deleted.

Comment 56: *Line 325: remove “specific enzymatic or non-enzymatic”*

Response: We have revised the discussion section. This sentence was removed.

Comment 57: *Line 431: Were the same 45 cycles also used? If so this is a lot of cycles that will introduce sequencing errors.*

Response: While 45 cycles were used in the PCR, the influence of cycle number is largely dependent on the method by which libraries are built. In this dataset we quantified the DNA in each extract by qPCR so as to ‘tune’ the level of input DNA. Then, rather than apply a common 2-step amplification (susceptible to chimera and tag-jumping) we use a 1-step library build that has large fusion primers. While there are many upside to single-step library the efficiency of the PCR is lowered. Accordingly, we typically increase the cycle number to ensure all reactions have plateaued. We remain confident that the single-step library and subsequence data filtering is more than sufficient to deal with any systematic influence of sequencing error.

Comment 58: *Line 467: “blasting” is not a word. Change to “determined by performing a BLAST search”*

Response: We have now changed “blasting” to “determined by performing a BLAST search” as suggested (Line 480-481).

Line 480-481: Fungal phylogeny was determined by performing a BLAST search of the obtained ITS sequences against the NCBI database.

Comment 59: *Line 523: Was a quality filter also used? This need to be added.*

Response: According to the Miseq SOP (https://www.mothur.org/wiki/MiSeq_SOP), sequences are filtered based on the quality score. This is done using the make.contigs command. We have now added the information (Line 577-579).

Line 577-579: According to the MiSeq standard operating procedure (https://www.mothur.org/wiki/MiSeq_SOP), sequences were filtered based on the quality score using the make.contigs command.

Comment 60: Line 524: Add the percent identity used to create the OTUs.

Response: We have now added the information (Line 579-580).

Line 579-580: Sequences were then trimmed to a length of 250 bp, screened for chimeras (UCHIME), and grouped into OTUs at 0.03 distance cut-offs.

References:

1. Peterson, P. J. & Minski, M. J. Precious Metals and Living Organisms. *Interdisciplinary Science Reviews* **10**, 159–169 (1985).
2. Kayano, Y., Tanaka, A., Akano, F., Scott, B. & Takemoto, D. Differential roles of NADPH oxidases and associated regulators in polarized growth, conidiation and hyphal fusion in the symbiotic fungus *Epichloë festucae*. *Fungal Genetics and Biology* **56**, 87–97 (2013).
3. Jean, G. E. & G. Michael, B. An XPS and SEM study of gold deposition at low temperatures on sulphide mineral surfaces: Concentration of gold by adsorption/reduction. *Geochimica et Cosmochimica Acta* **49**, 979–987 (1985).
4. Vermeesch, P. Lies, Damned Lies, and Statistics (in Geology). *Eos, Transactions American Geophysical Union* **90**, 443–443 (2009).
5. Dahiru, T. P – VALUE, A TRUE TEST OF STATISTICAL SIGNIFICANCE? A CAUTIONARY NOTE. *Ann Ib Postgrad Med* **6**, 21–26 (2008).
6. Yasmin, A., Ramesh, K. & Rajeshkumar, S. Optimization and stabilization of gold nanoparticles by using herbal plant extract with microwave heating. *Nano Converg* **1**, (2014).
7. Trevors, J. T. Sterilization and inhibition of microbial activity in soil. *Journal of Microbiological Methods* **26**, 53–59 (1996).

8. Wolf, D. C., Dao, T. H., Scott, H. D. & Lavy, T. L. Influence of Sterilization Methods on Selected Soil Microbiological, Physical, and Chemical Properties. *Journal of Environmental Quality* **18**, 39–44 (1989).
9. Berns, A. E. *et al.* Effect of gamma-sterilization and autoclaving on soil organic matter structure as studied by solid state NMR, UV and fluorescence spectroscopy. *European Journal of Soil Science* **59**, 540–550 (2008).
10. Reith, F., Lengke, M. F., Falconer, D., Craw, D. & Southam, G. The geomicrobiology of gold. *ISME J* **1**, 567–584 (2007).
11. Bank, T. L. *et al.* Effects of gamma-sterilization on the physico-chemical properties of natural sediments. *Chemical Geology* **251**, 1–7 (2008).
12. Navalon, S., de Miguel, M., Martin, R., Alvaro, M. & Garcia, H. Enhancement of the Catalytic Activity of Supported Gold Nanoparticles for the Fenton Reaction by Light. *J. Am. Chem. Soc.* **133**, 2218–2226 (2011).
13. Bollen, G. J. The selective effect of heat treatment on the microflora of a greenhouse soil. *Netherlands Journal of Plant Pathology* **75**, 157–163 (1969).
14. Smit, E., Leeflang, P., Glandorf, B., Dirk van Elsas, J. & Wernars, K. Analysis of Fungal Diversity in the Wheat Rhizosphere by Sequencing of Cloned PCR-Amplified Genes Encoding 18S rRNA and Temperature Gradient Gel Electrophoresis. *Appl Environ Microbiol* **65**, 2614–2621 (1999).
15. Eggleton, R. A. *The regolith glossary: Surficial geology, soils, and landscapes*. (Cooperative Research Centre for Landscape Evolution and Mineral Exploration, 2001).
16. Kristensen, E. *et al.* What is bioturbation? The need for a precise definition for fauna in aquatic sciences. *Marine Ecology Progress Series* **446**, 285–302 (2012).
17. Anand, R. R., Aspandiar, M. F. & Noble, R. R. P. A review of metal transfer mechanisms through transported cover with emphasis on the vadose zone within the Australian regolith. *Ore Geology Reviews* **73, Part 3**, 394–416 (2016).

18. Reith, F. & McPhail, D. C. Effect of resident microbiota on the solubilization of gold in soil from the Tomakin Park Gold Mine, New South Wales, Australia. *Geochimica et Cosmochimica Acta* **70**, 1421–1438 (2006).
19. Southam, G., Lengke, M. F., Fairbrother, L. & Reith, F. The Biogeochemistry of Gold. *Elements* **5**, 303–307 (2009).
20. Bhattarai, A., Bhattarai, B., P, S. & ey. Variation of Soil Microbial Population in Different Soil Horizons. *Journal of Microbiology & Experimentation* **2**, 1–0 (2015).
21. Robb, F., Davies, B. R., Cross, R., Kenyon, C. & Howard-Williams, C. Cellulolytic bacteria as primary colonizers of *Potamogeton pectinatus* L. (Sago Pond Weed) from a Brackish South-Temperate Coastal Lake. *Microb Ecol* **5**, 167–177 (1979).
22. Jasrotia, P. *et al.* Watershed-Scale Fungal Community Characterization along a pH Gradient in a Subsurface Environment Cocontaminated with Uranium and Nitrate. *Appl. Environ. Microbiol.* **80**, 1810–1820 (2014).
23. Smith, M. D., Hartnett, D. C. & Rice, C. W. Effects of long-term fungicide applications on microbial properties in tallgrass prairie soil. *Soil Biology and Biochemistry* **32**, 935–946 (2000).

REVIEWERS' COMMENTS:

Reviewer #1 (Remarks to the Author):

I commented on an earlier version of this manuscript. I am generally satisfied with the revisions and the replies to my comments. Especially, the presentation of the geochemical data is improved. I have only a few relatively minor remarks on the current versions. I hope the authors will consider these remarks in order to make the manuscript more suitable for publication.

1. Introduction, lines 47-50: "fossilized fungi involved in the geological process and distinguish their past activities from those of bacteria or archaea¹⁴" Here it could also be worth to mention that fungi are usually better preserved through mineralization than prokaryotes which further make it difficult to differentiate between the importance of their respective activities, and add reference "Drake H, Ivarsson M, Bengtson S, Heim C, Siljeström S, Whitehouse MJ, Broman C, Belivanova V, Åström ME. Anaerobic consortia of fungi and sulfate reducing bacteria in deep granite fractures. Nature Communications 2017;8(1):55."

2. Introduction, lines 51-52. Sentence: "The carbon seams were originally proposed to represent remnants of eukaryotic microorganisms¹⁶" does not seem to be fully correct. The cited reference actually rules out influence of microbial mats in the metal accumulation process. Furthermore, the previous models of accumulation in association with microbial mats mostly deal with cyanobacteria (Frimmel and Hennigh, 2015), which are prokaryotes, so I would not say that models including eukaryotes are general.

3. Lines 53: "Erosion of a gold sheet" is very unspecific, please write out more details, e.g. that the study was about mobilization of Au from electronic waste by fungi, or similar.

4. Fig. 2, I am not fully convinced that a bar diagram with standard deviation is the best way to show Au concentration (n=3) of three selected values that are very varied (maybe addition of three symbols would be informative to this particular case). In addition, it would be interesting to show the "non-hotspot" values in the gold anomaly as bars as well. If they are more similar to the hotspot values for all variables or more similar to the reference. i.e. to make three bars for each panel.

5. Line 151: HSO₄⁻ should be HSO₄⁻

6. Fig. 5. Include in caption for "c" the spot size, i.e. that not only Au particles are included in the measurement but also the surrounding material. I know that the authors have given a sufficient reply to my earlier comment on this, but still I recommend addition of this information to the caption.

7. Fig. S1, please make plot for gold as Log-scale so that the median and box can be compared. Now the large anomaly caused by a single sample makes comparisons impossible (both squeezed in at the bottom of the graph).

8. Fig. S5 should be put in the main text, and please guide the reader through the steps (arrows) with text in the caption. It is not straightforward where the process begins and ends.

Reviewer #2 (Remarks to the Author):

I feel that the points raised in the previous round of review have been satisfactorily addressed by the authors and the current revised version of the manuscript entitled "Evidence for fungi and gold redox interaction under Earth surface conditions" by Dr Bohu and colleagues can be published in Nature Communications.

Reviewer #3 (Remarks to the Author):

This is a much improved manuscript and I thank the authors for a thorough explanation for on each reviewers comments.

I disagree with the number of PCR cycles that was used to amplify the DNA because of polymerase caused mutations and thermal degradation that happens over that long time period. For example, see the references Potapov V, Ong JL (2017) Correction: Examining Sources of Error in PCR by Single-Molecule Sequencing. PLOS ONE 12(7): e0181128 and E. Pienaar, M. Theron, M. Nelson, H.J. Viljoen, (2006) A quantitative model of error accumulation during PCR amplification, Computational Biology and Chemistry, 30(2). However, I agree in this instance the errors introduced do not have a significant affect on the major conclusions. I, therefore, don't have any other comments that need to be addressed.

Response to Comments from Reviewers

We sincerely thank the reviewers for their constructive comments. Below is a new point-to-point response to the specific concerns after the last round. We have repeated the comments verbatim in italics. The revised content was highlighted in blue color.

Reviewer #1 (Remarks to the Author):

Comment 1: *I commented on an earlier version of this manuscript. I am generally satisfied with the revisions and the replies to my comments. Especially, the presentation of the geochemical data is improved. I have only a few relatively minor remarks on the current versions. I hope the authors will consider these remarks in order to make the manuscript more suitable for publication.*

Response: We thank reviewer 1 for the encouraging comment. We have now revised the manuscript according to the suggestions.

Comment 2: *Introduction, lines 47-50: “fossilized fungi involved in the geological process and distinguish their past activities from those of bacteria or archaea¹⁴” Here it could also be worth to mention that fungi are usually better preserved through mineralization than prokaryotes which further make it difficult to differentiate between the importance of their respective activities, and add reference “Drake H, Ivarsson M, Bengtson S, Heim C, Siljeström S, Whitehouse MJ, Broman C, Belivanova V, Åström ME. Anaerobic consortia of fungi and sulfate reducing bacteria in deep granite fractures. Nature Communications 2017;8(1):55.”*

Response: We have now added the text and the reference (Line 51-53).

Line 51-53: In addition, fungi are usually better preserved through mineralization than prokaryotes, which further makes it difficult to differentiate between the importance of their respective activities¹⁵.

Comment 3: *Introduction, lines 51-52. Sentence: “The carbon seams were originally proposed to represent remnants of eukaryotic microorganisms¹⁶” does not seem to be fully correct. The cited reference actually rules out influence of microbial mats in the metal accumulation process. Furthermore, the previous models of accumulation in association with microbial mats mostly deal with cyanobacteria (Frimmel and Hennigh, 2015), which are prokaryotes, so I would not say that models including eukaryotes are general.*

Response: We have deleted the reference and the related sentences.

Comment 4: *Lines 53: “Erosion of a gold sheet” is very unspecific, please write out more details, e.g. that the study was about mobilization of Au from electronic waste by fungi, or similar.*

Response: Many thanks for the suggestion. We have now revised the sentence (Line 55-56).

Line 55-56: Recently, fungi have been found to mobilize gold from electronic waste¹⁸, suggesting they can interact with metallic gold, possibly through biological redox transformations.

Comment 5: *Fig. 2, I am not fully convinced that a bar diagram with standard deviation is the best way to show Au concentration (n=3) of three selected values that are very varied (maybe addition of three symbols would be informative to this particular case). In addition, it would be interesting to show the “non-hotspot” values in the gold anomaly as bars as well. If they are more similar to the hotspot values for all variables or more similar to the reference. i.e. to make three bars for each panel.*

Response: In the gold anomalous area, we have total ten sampling sites. Three out of the ten sites were identified as hotspots because the *in situ* gold concentrations of these sites were 1.5-fold greater than or equal to the median (3.54 ng g⁻¹). It does not make sense to us to add three other symbols with lower gold content to represent the *in situ* gold concentrations of the hotspots. Nevertheless, we agree with Reviewer

1 that we have now revised Figure 2 to four bars including the data of the hotspots, the non-hotspots, all spots of the gold anomaly, and the data of the reference area. Thus, the comparison of the geochemical data between the auriferous and reference area should be clear. We have now combined the former Figure 2 and Supplementary Figure 1 to Figure 1.

Figure 1. Soil geochemical parameters of the gold anomaly and the adjacent reference area. The bars H, N, and A represent the data of the hotspots, the non-hotspots, and all spots of the gold anomaly, respectively. The significance of the differences between two sets of data was determined by an unpaired *t*-test (GraphPad Prism Version 7). The *P* value for gold content was based on a one-tailed *t*-test calculation. $P < 0.05$ was considered statistically significant. The error bar indicates the standard deviation.

Comment 6: Line 151: HSO_4^- should be HSO_4^-

Response: Fixed.

Comment 7: *Fig. 5. Include in caption for “c” the spot size, i.e. that not only Au particles are included in the measurement but also the surrounding material. I know that the authors have given a sufficient reply to my earlier comment on this, but still I recommend addition of this information to the caption.*

Response: We have now revised the caption of panel c (Line 942-944).

Line 942-944: (c) Detailed view of nanometer gold particles and surrounding materials (yellow arrows) over the surface of fungal hyphae. The inset shows an EDS profile of the major components of the assemblage.

Comment 8: *Fig. S1, please make plot for gold as Log-scale so that the median and box can be compared. Now the large anomaly caused by a single sample makes comparisons impossible (both squeezed in at the bottom of the graph).*

Response: We have now changed the format of the former Supplementary Figure 1 and combined it to Figure 2. The new figure was shown as Figure 1. Please also see Response for Comment 5.

Comment 9: *Fig. S5 should be put in the main text, and please guide the reader through the steps (arrows) with text in the caption. It is not straightforward where the process begins and ends.*

Response: Many thanks for the constructive comment. We have now put the former Fig. S5 in the main text as Figure 7. The caption has now been revised to explain the direction of the process (Line 959-962).

Line 959-962: The conceptual model for mycological gold redox transformation under Earth surface conditions. Superoxides from fungal hyphae oxidatively dissolve colloidal gold to gold ions with likely assistance of protons. Gold ions then complex with the intracellularly produced ligand. Colloidal gold

nanoparticles may be regenerated from the interaction between gold complexes and reduced organic carbon species. Dashed arrows indicate superoxides and ligands are produced intracellularly.

Reviewer #2 (Remarks to the Author):

Comment 10: *I feel that the points raised in the previous round of review have been satisfactorily addressed by the authors and the current revised version of the manuscript entitled "Evidence for fungi and gold redox interaction under Earth surface conditions" by Dr Bohu and colleagues can be published in Nature Communications.*

Response: We thank Reviewer 2 for the efforts and valuable comments to our manuscript.

Reviewer #3 (Remarks to the Author):

Comment 11: *This is a much improved manuscript and I thank the authors for a thorough explanation for on each reviewers comments.*

I disagree with the number of PCR cycles that was used to amplify the DNA because of polymerase caused mutations and thermal degradation that happens over that long time period. For example, see the references Potapov V, Ong JL (2017) Correction: Examining Sources of Error in PCR by Single-Molecule Sequencing. PLOS ONE 12(7): e0181128 and E. Pienaar, M. Theron, M. Nelson, H.J. Viljoen, (2006) A quantitative model of error accumulation during PCR amplification, Computational Biology and Chemistry, 30(2). However, I agree in this instance the errors introduced do not have a significant affect on the major conclusions. I, therefore, don't have any other comments that need to be addressed.

Response: We thank Reviewer 3 for the efforts and valuable comments to our manuscript.